# When Will First-Price Work Well? The Impact of Anti-Corruption Rules on Photovoltaic Power Generation Procurement Auctions

**Peng Hao [1],[*],[†],[‡]** , **Jun-Peng Guo [2],[‡]**, **Eoghan O'Neill [3],[4],[‡]** and **Yong-Heng Shi [2],[‡]**

[1] College of Economics and Management, Tianjin Renai College, Tianjin 301636, China
[2] College of Management and Economics, Tianjin University, Tianjin 300072, China
[3] Energy Policy Research Group (EPRG), University of Cambridge, Cambridge CB2 1AG, UK
[4] Econometric Institute, Erasmus School of Economics, Erasmus University Rotterdam, 3062 Rotterdam, The Netherlands
[*] Correspondence: phao@tju.edu.cn; Tel.: +86-1852-238-3031
[†] Current address: Tuanboxincheng, Boxueyuan, Jinghai District, Tianjin 301636, China.
[‡] These authors contributed equally to this work.

**Abstract:** Along with the prevalence of photovoltaic (PV) procurement contracts, the corruption between auctioneers and potential electricity suppliers has attracted the attention of energy regulators. This study considers a corruption-proof environment wherein corruption is strictly suppressed. It elaborates a mechanism to explore the impact of corruption-proof measures on PV procurement auctions. It adopts incentive compatible constraints based on revelation principle to reflect PV firms' optimal utilities. It employs first-price and first-score auctions and uses the Bayesian Nash equilibrium to provide a description of market outcomes. The results show that several strategies have different impacts on social welfare, PV firms' utility, and the benefits of corruption. First, a first-price auction cannot act as a suitable policy because it may encourage corruption. Second, the first-score choice is desirable for social welfare to fit the forthcoming high-quality and low-price surroundings. Third, the first-score strategy maximizes PV firms' utility and total income. The implications suggest that regulators ought not to employ first-price auctions in the future PV market from the perspective of social welfare. Another disadvantage of the first-price approach is that it enables the PV firm to maintain the utmost benefit from corruption.

**Keywords:** photovoltaic procurement auction; corruption-proof measure; first-price; first-score

## 1. Introduction

Corruption is a corrosive force that eviscerates the vitality of businesses and stunts a country's economic potential. Public procurement, a crucial way of implementing government budgets, can be highly vulnerable to corruption [1]. Bidding corruption is extensively distributed in the current procurement of solar photovoltaic (PV) plant projects. Notably, 61% of the PV projects in Serbia [2] is impeded by corruption. Approximately 28% of Indian PV projects are forcefully corrupt-laden at the contracting stage [3]. An increase in public subsidies for PV spawned a relational increase in corruption activity across 76 Italian provinces [4]. Concerns have also been raised that large-scale PV projects in Bangladesh [5], Morocco [6], Tanzania [7], and sub-Saharan Africa [8] would be susceptible to corruption. The Republic of Korea's JoongAng Ilbo reported on 14 September 2022, that out of 261.6 billion won in questionable funds, 210.8 billion won, or 80.5%, was related to solar energy projects [9]. Bidding corruption in PV procurement is also a concern for regulators and is extensively distributed in China under the trend of large-scale PV [10].

There are two type of corruption, vertical corruption and horizontal corruption. A vertical corrupt practice is offering, giving, receiving, or soliciting, directly or indirectly, anything to influence improperly the actions of another party [11]. Corruption would



never be a concern if the principal, for example, the buyer, could make direct procurements from PV suppliers without leaving any leeway to the auctioneer, who is in charge of the procurement process [12]. In this context, a corrupt PV auction refers to a settlement through which the PV bidder transfers money to the auctioneer to maximize its payoff during the PV engineering procurement process. The other widely prevalent form of a collusive bid is bid rigging, namely, horizontal corruption. It arises when a subset, or possibly all, of the bidders, acts collusively and engages in bid rigging with a view to obtaining higher prices [13] in a procurement auction.

The corruption of PV procurement auctions practically appears to be vertical. The reasons for this are presented as follows: first, PV procurement is currently in a standard competition market because the generation costs of PV firms decline synchronously and almost approximate to each other [14], along with the PV subsidy on power generation withdrawals [15–17]. This limits PV bidders' profits. Consequently, the winner cannot afford to pay members within the coalition. For example, a PV firm pays vertical corruption 1 (USD) to the auctioneer or horizontal corruption 1 (USD) to 10 bidders within a coalition, which will spend 10 (USD), and receives the same winning result. If the bidder can guarantee a sufficient profit, he will adopt both vertical and horizontal corruption; otherwise, he will rather adopt the cheaper one. Second, the openness of bidding makes bid rigging impossible. Notably, PV auctions are a type of bidding open to foreign potential contractors. This captures the case when a firm comes from a country where the corruption of foreign civil servants is severely prosecuted, for instance, in the United States [18]. For these two reasons, bid rigging among PV bidders is hardly sustainable.

The present paper focuses on corruption in multidimensional auctions. First, we elaborate on the *first-price* procurement auction, that is, a one-dimensional scenario in which only price matters. Here, we exclude price manipulation actions that are unsustainable vis-à-vis external oversight [12]. Second, we deal with the *first-score* case, in which quality and price matter. This is because the PV generator's qualitative characteristics, such as operation and maintenance, modules and panels, and voltage stability [19–21], are vital for the principal. The auctioneer in charge of evaluating proposals can choose to be corrupt by manipulating the quality assessment to favor a bribing supplier [12,22,23]. Scoring auctions exploring the implementation of multidimensional auctions and the properties of optimal mechanisms [24–28] are a prevalent tool for bidding evaluation.

Our study is novel in many respects. First, to suppress corruption, we consider hidden information and design mechanisms to screen the asymmetric information held by bidders. This differs from some prior research on corruption [29,30] that involve general equilibrium as a concern. We employ the revelation principle and add incentive compatibility (IC) and individual rationality (IR) constraints to solve a Bayesian Nash equilibrium issue. Second, we find that the first-score strategy performs well in improving social welfare in forthcoming market conditions.

The Bayesian Nash equilibrium is adopted as research topic and the principle of revelation is adopted as method to solve this problem. The reasons are as follows. Firstly, the Nash equilibrium solves the problem of game, while the general equilibrium deals with the optimization problem. The game problem in the Nash equilibrium is to derive an equilibrium provided the strategies of the two players. The objective function in the Nash equilibrium and expressed in terms of the player's optimal utility, requires an implicit variable. That is the key difference between the Nash equilibrium and the general equilibrium. Also, the Nash Equilibrium reflects the optimization of PV power users and PV suppliers. This problem is also a Bayesian problem because a discrete optimization involves two randomly selected actors. Secondly, the adoption of the revelation principle solves the Bayesian Nash equilibrium, because it can make PV enterprises of true types. In this way, PV enterprises can get the best utilities. To this end, we add IC constraints to make PV enterprises to tell the truth, and IR constraints to ensure that PV enterprises participate in the game.

The remainder of this paper is organized as follows: Section 2 reviews the issues related to corruption. Section 3 presents the modeling assumptions and scenarios and develops the main mathematical model. Section 4 simulates the model in connection to equilibrium. Section 5 discusses the contribution of theoretical and simulation analysis and presents policy implications. Section 6 summarizes the novelty of this study.

## 2. Related Literature

A strand of the literature studies corruption in multidimensional procurement auctions [31], whereby the government may care about both the price and quality of the project. It is worth noting that the problem of quality manipulation arises when the auctioneer distorts the reports of bid quality scores. Burguet and Che [23] consider a two-bidder auction where the agent compares corruptions $b_1$ and $b_2$. If $b_1 > b_2$, then the agent favors Firm 1 by enlarging its quality by a multiplier $m$, as long as Firm 1 wins with the manipulation. Celentani and Ganuza [22] set a case where the agent is randomly matched to one firm and demands corruption in exchange for the agreement that the firm will be awarded the project and permitted to produce lower quality.

Among the recent first-score-auction-related studies, Wang [12], Burguet [29], and Huang and Xia [32] are closely related to this paper. The first and second studies investigate the design of procurement mechanisms. We differ from the first study in that we design a unilateral control from the perspective of a regulator to curtail corruption. However, Wang [12] analyzes the features of existent external oversight. We differ from Burguet [29] in that we use the revelation principle to let PV bidders tell their truth types under the IC constraint. The insightful paper by Burguet [29] lacks a revelation principle that applies to this case. Huang and Xia [32] is also related to our research in that they consider a favoritism auction. We consider an endogenous favoritism arrangement case in which the auctioneer does not definitely favor any firm as they submit contract bids. However, Huang and Xia [32] considers an exogenous favoritism auction where the inefficient firm is always corrupt. The advantage of exogenous favoritism is that it reflects the reality of procurement auctions. In PV procurement auction practice, the auctioneer does not favor any company in the bidding process. The winning bidder left part of the corrupt proceeds to the auctioneer as a reward. However, the auctioneer could not distinguish the winner beforehand. Therefore, we use exogenous favoritism to study the establishment of the models.

It is widely observed that there is considerable research on corruption in the power sector. Klemperer [33] first states that corruption is hard in a bidding scheme in the electricity market. Dechenaux and Kovenock [34] conducts a similar study and sustains that result. Recent studies have drawn opposite conclusions. This stream includes, but is not limited to [35], who finds that the bidder shows a high probability of joining a coalition. Samadi and Hajiabadi [36] evaluates the possibility of coalition formation in the power market. Palacio [37] predicts corruption patterns in a liberalized electricity market with mandatory auctions of forward contracts and finds that the price increases in this surrounding. Nevertheless, studies on vertical corruption in power generation have rarely addressed the PV industry.

The exploration of new procurement methods has become a popular approach toward reducing power generation corruption. Woo et al. [38] first study the electricity procurement cost and risk control in Internet-based multi-round auctions. This line of research also includes studies introducing competing bidders into the procurement of electricity supply contracts to mitigate tacit corruption [39]. Studies on the design of an auction mechanism to prevent corruption have been extended to asymmetric information. In a study on unilateral payment behavior, Che et al. [40] shows that an interesting feature of the optimal mechanism is the asymmetric treatment of bidders who are ex-ante identical. Although this strand of the literature has been significantly expanded by the analysis of mechanism design, research on anti-corruption policy options in different markets has not been sufficiently thorough.

The existing literature offers the following insights: first, price manipulation in first-price auctions is difficult to maintain under strict supervision. Second, quality manipulation is prevalent and well-distributed among recent corrupt studies in first-score auctions. Third, research on power generation rarely deals with corruption in PV procurement auctions. Fourth, policies to curb PV corruption require further research to adapt to different markets.

This study focuses on the influence of corruption-proof measures on PV procurement auctions. The motivations behind this research are twofold. First, corruption-proof measures are prevalent along with the increase in PV corruption. Strict external supervision includes but is not limited to China's recent strong supervision of the PV power generation industry and PV power station construction [41]. Second, the potential sub-optimality of the first-price auction leading to the dysfunction has been gradually recognized [42,43]. Therefore, this study addresses the following issues:

Q1. What are the advantages and disadvantages of the first-price concept for PV procurement auctions under a strict corruption-proof environment?

Q2. What strategy, first-price or first-score, should regulators employ to maximize social welfare?

## 3. The Model

### 3.1. Assumptions and Scenarios

The business model involves three participants: PV power generation firms, users, and auctioneers. The auctioneer receives remuneration and is responsible for the project auction. Notably, PV firms engage in corrupt practices with auctioneers to win. The winner provides power to the users. Figure 1 illustrates an overview of the PV scheme.

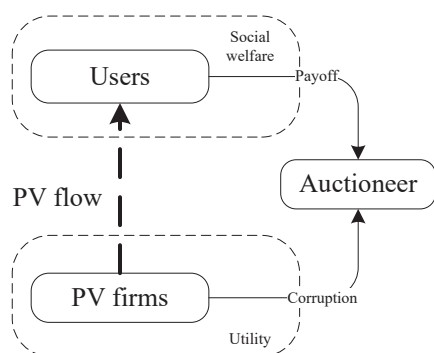

**Figure 1.** Business model of a PV procurement auction.

This model is more complicated than that associated with fossil energy generation procurement. Accordingly, PV procurement must deal with the corruption-proofing of bidding that a normal fossil generation procurement rarely entails. This is mainly because the grid procures electricity from coal-fired power using a dispatch schedule other than bidding.

**Assumption 1.** *Bidders 1 and 2 are randomly selected from all PV bidders. Let $P_i$ $(i = 1, 2)$ be their announced prices. $p_h$ and $p_l$ denote the high and low prices of Bidders 1 and 2, respectively. Let $\alpha \triangleq \text{Prob}\,(P_i = p_l)$ and let $P$ be $\Delta p = p_h - p_l$, where $\Delta p$ is positive.*

**Assumption 2.** *The auctioneer has a probability of $x_i(P_1, P_2) \in [0, 1]$ to choose PV firm i, where $x_1 + x_2 \equiv 1$. The tax rate and the cost of internal transfer from the winner to the auctioneer are denoted by $\lambda$ and $\lambda_f$. A tax of one unit will bring a monetary burden $\frac{1}{1-\lambda}$ to taxpayers. A transfer of one unit will result in $\frac{1}{1+\lambda_f}$ to auctioneer.*

**Assumption 3.** *PV firm i's cost $C_i$ is defined as $C_i(P_1, P_2) = P_i - c_i(P_1, P_2)$, where $c_i$ is PV firm i's benefit obtained from corruption. $U_i(p_j) = t_i - x_i(P_1, P_2)\psi(c_i(P_1, P_2))$ refers to the expected*

utility of Firm $i$ with price type $j$. $U_i(p_j)$ is realized by the difference in the net transfer $t_i$ and the negative effect of corruption $\psi(c_i)$ with probability $x_i(P_1, P_2)$, $i = 1, 2$, $j = h, l$. PV firms' profits come from two parts. One is $i$'s benefit from corruption $c_i(P_1, P_2)$ and the other is $i$'s utility $U_i(p_j)$.

$C_i$ and $c_i$ are endogenous variables related to the announced prices $P_i$. $c_i$ depends on both bidders' prices, $P_1$ and $P_2$. Consider the first-price as an example. The two PV firms have the same cost $C_i(P_1, P_2) = 50$ (USD/MWh) and different announced prices $P_1 = 60$ (USD/MWh) and $P_2 = 66$ (USD/MWh). Here, Bidder 1 is the winner and $c_1(P_1, P_2) = 10$ (USD/MWh). If $P_2 = 59$ (USD/MWh), then Bidder 2 wins and Bidder 1 loses, $c_1(P_1, P_2) = 0$ (USD/MWh). Therefore, Bidder 1's benefits from corruption switch at both bidders' prices. This is also true for the first-score case. Net transfer payments $t_i$ given to PV firms come from free land, goodwill earned from implementing projects, and surplus from project payments after cost offset. A risk-averse generator has a numerical expression for the disutility of corruption, that is, $\psi(c_i)$, which satisfies $\psi' \geqslant 0$ and $\psi'' \geqslant 0$. These convex conditions agree with the assumptions of [44–46].

This study explores variations in analytical models across the following different sets of scenarios. First price (FP) involves a fairly restrictive scheme that excludes high prices, leading to $x_1^{FP}(p_h, P_2) = x_2^{FP}(P_1, p_h) = 0$. First score (FS) is a multidimensional scheme in which price and quality are important. The variables and parameters used in this study are defined in Table 1.

**Table 1.** Nomenclature.

|  | Parameters & Variables | Notations |
|---|---|---|
| Parameters | $Q_i$ | Quality of PV firm $i$ (USD/MWh) |
|  | $R$ | Auctioneer's remuneration (USD/MWh) |
|  | $P_i$ | Price of PV firm $i$ (USD/MWh) ($i = 1, 2$) |
|  | $p_j$ | Price type $j$ (USD/MWh) ($j = h, l$) |
|  | $\lambda$ | Tax rate (%) |
|  | $p$ | Probability on $P_i = p_l$ (dimensionless) |
|  | $k$ | Rate of corruption's dis-utility (dimensionless) |
|  | $\delta$ | Probability of information states (dimensionless) |
|  | $\lambda_f$ | Cost of secret transfer to auctioneer (%) |
|  | $\alpha$ | Probability of low price (%) |
| Variables | $C_i(P_1, P_2)$ | Cost of of PV firm $i$ (USD/MWh) ($i = 1, 2$) |
|  | $x_i(P_1, P_2)$ | Probability of auctioneer choosing PV firm $i$ (dimensionless) ($i = 1, 2$) |
|  | $c_i(P_1, P_2)$ | Benefit from corruption of PV firm $i$ (USD/MWh) ($i = 1, 2$) |
|  | $\psi(c_i(P_1, P_2))$ | PV firm $i$'s dis-utility of corruption (USD/MWh) ($i = 1, 2$) |
|  | $t_i(P_1, P_2)$ | Net transfer to PV firm $i$ (USD/MWh)($i = 1, 2$) |
|  | $U_i(p_j)$ | Utility of PV firm $i$ with price type $j$ (USD/MWh) ($i = 1, 2; j = h, l$) |
|  | $R_{s_1}, R_{s_2}$ | Auctioneer's remuneration at state 1 and 2 (USD/MWh) |
|  | $S$ | Social welfare (USD/MWh) |

### 3.2. Model Setup

The model aims to maximize users' surplus $S$, which is defined as the difference between users' willingness to pay, that is, quality $Q$, and their actual payment. According to many classical studies [47,48], actual payment is equivalent to a cost reimbursement discipline where users hedge cost $C_i$ with probability $x_i$ and deliver net transfer $t_i$ to the

winning PV generator. We allow interaction between the auctioneer and users, where the auctioneer organizes a bid on behalf of users, and users transfer $R$ to the auctioneer as a remuneration. Therefore, the users' surplus $S$ can be represented as

$$S(Q, P) = Q - \frac{1}{1 - \lambda} \left\{ \mathbb{E}_{P_i}[x_i(P_1, P_2)C_i(P_1, P_2) + t_i(P_1, P_2)] + R \right\}, \tag{1}$$

where $\mathbb{E}_{P_i}[\cdot]$ denotes the expectation when $P_1$ and $P_2$ take the value of $p_h$ or $p_l$, respectively. By replacing $C_i(P_1, P_2)$ and $t_i(P_1, P_2)$ with their equivalents $P_i - c_i(P_1, P_2)$ and $U_i(p_j) + x_i(P_1, P_2)\psi(c_i(P_1, P_2))$, the users' surplus has the following expression

$$S(Q, P) = Q - \frac{1}{1 - \lambda} \left\{ \mathbb{E}_{P_i}[x_i(P_1, P_2)(P_i - c_i(P_1, P_2) + \psi(c_i(P_1, P_2)))] + \mathbb{E}_{p_j}[U_i(p_j)] + R \right\}, \tag{2}$$

To reveal PV firms' utilities $U_i(p_j)$, $i = 1, 2; j = h, l$, our regulatory scheme derives their maximum by letting them tell the truth. The reasons for this are as follows: first, to win, one bidder announces a lower private price $p_j$ than its true productive efficiency type. However, such behavior causes the bidder to suffer the danger of a delay in the completion of the project. Second, the bidder can also offer a higher price, which can cause him to lose the contract. As a result, bidders must offer their true prices. At this time, the PV firm has no incentive to become corrupt. Telling the truth is a strict regulation for PV procurement auctions. We treat the announced prices as PV firms' asymmetric information from the regulator's perspective.

To induce bidders to tell the truth, we employ the revelation principle [49] to screen this asymmetric information in the scenarios discussed hereinafter and draw some important findings in Lemma 1.

**Lemma 1.** *The utility of a high-price firm is fixed at the retained revenue only, that is,* $U_1(p_h) = U_2(p_h) = 0$; *the low-price firm receives the expected utilities of* $U_1(p_l) = \mathbb{E}_{P_2}[x_1(p_h, P_2)\Phi(c_1(p_h, P_2))]$, $U_2(p_l) = \mathbb{E}_{P_1}[x_2(P_1, p_h)\Phi(c_2(P_1, p_h))]$, *where* $\Phi(c) = \psi(c) - \psi(c - \Delta p)$.

Lemma 1 leads to $U_1(p_l) = \alpha x_1(p_h, p_l)\Phi(c_1(p_h, p_l)) + (1 - \alpha)x_1(p_h, p_h)\Phi(c_1(p_h, p_h))$ and $U_2(p_l) = \alpha x_2(p_l, p_h)\Phi(c_2(p_l, p_h)) + (1 - \alpha)x_2(p_h, p_h)\Phi(c_2(p_h, p_h))$. Based on Lemma 1, we obtain $\mathbb{E}_{p_j}[U_1(p_j)] = \alpha U_1(p_l) + (1 - \alpha)U_1(p_h) = \alpha U_1(p_l)$ and $\mathbb{E}_{p_j}[U_2(p_j)] = \alpha U_2(p_l) + (1 - \alpha)U_2(p_h) = \alpha U_2(p_l)$. Appendix A.1 attaches the proof of Lemma 1.

### 3.3. First Price

The nature of the first-price concept is as follows. First, a PV firm offering a high price is not awarded the contract. Second, $Q$ is constant because quality is deemed homogeneous, and only price matters. Third, price manipulation does not occur because the external oversight curtails it [12]. Fourth, the PV firm has the incentive to become corrupt because it is unsure whether it is a low-price firm. The timeline of a first-price procurement auction is shown in Figure 2. The regulator proposes a first-price bidding rule, and PV firms approach the auctioneer. The auctioneer does nothing under external oversight. Moreover, PV firms submit their price bids, while the winner benefits from corruption and transfers a part to the auctioneer as a trade-off.

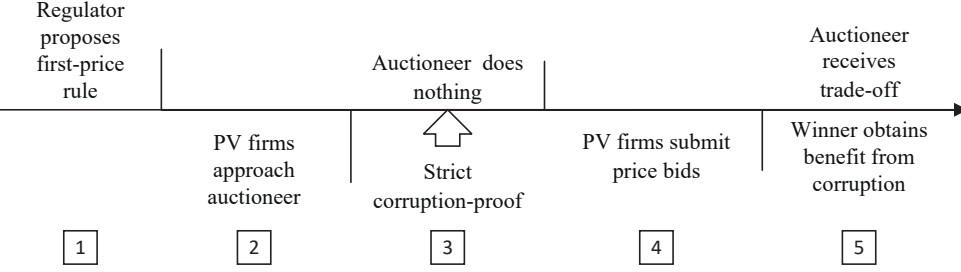

**Figure 2.** Timeline of first-price.

Model (3) describes the following features. (3a) determines PV firm $i$'s chance of winning and its benefit from corruption, where

$$S(P) = Q - \frac{1}{1-\lambda}\left\{\mathbb{E}_{P_i}[x_1(P_1,P_2)(P_1 - c_1(P_1,P_2) + \psi(c_1(P_1,P_2)))] + \mathbb{E}_{p_j}[U_1(p_j)]\right\}$$
$$- \frac{1}{1-\lambda}\left\{\mathbb{E}_{P_i}[x_2(P_1,P_2)(P_2 - c_2(P_1,P_2) + \psi(c_2(P_1,P_2)))] + \mathbb{E}_{p_j}[U_2(p_j)]\right\} - \frac{R}{1-\lambda}, j = h, l.$$

Constraints $IC_1$ and $IC_2$ determine $i$'s utility. $IR_1$ is naturally met because $x_i(P_1,P_2) \geqslant 0$, $\Phi(c) = \psi(c) - \psi(c - \Delta p)$, $\psi' \geqslant 0$. (3b) excludes high prices. Proposition 1 summarizes our conclusions.

$$\max_{x_i^{FP}(\cdot,\cdot), c_i^{FP}(\cdot,\cdot)} S^{FP} = S(P) \tag{3a}$$

$$s.t. \begin{cases} U_1(p_l) = \mathbb{E}_{P_2}[x_1(p_h,P_2)\Phi(c_1(p_h,P_2))] : (IC_1) \\ U_2(p_l) = \mathbb{E}_{P_1}[x_2(P_1,p_h)\Phi(c_2(P_1,p_h))] : (IC_2) \\ U_i(p_j) \geqslant 0 : (IR_1), i = 1,2; j = h,l \\ \sum_{i=1}^{2} x_i(P_1,P_2) = 1, x_i(P_1,P_2) \in [0,1] \\ x_1(p_h,P_2) = x_2(P_1,p_h) = 0 \end{cases} \tag{3b}$$

**Proposition 1.** *The first-price strategy brings the winner benefit from corruption $c^*$ or $\tilde{c}$ where $\psi'(c^*) = 1$, $\psi'(\tilde{c}) = 1 - \frac{\alpha}{1-\alpha}\Phi'(\tilde{c})$, $\Phi(c) = \psi(c) - \psi(c - \Delta p)$; first-price gives utilities $U_1(p_l) = \frac{1}{2}(1-\alpha)\Phi(c_1(p_h,p_h))$ and $U_2(p_l) = \frac{1}{2}(1-\alpha)\Phi(c_2(p_h,p_h))$ to the low-price PV firm and only leaves high-price PV firm retained utility, that is, $U_i^{FP}(p_h) = 0$.*

Proposition 1 shows that both firms can benefit from corruption within the first-price strategy. See Appendix B.1 for further details. We can conclude that the winner obtains benefits from corruption $c_i^{FP}(P_1,P_2)$ equal to $c^*$ or $\tilde{c}$, while the loser receives 0. The first-price concept leaves a high-price firm reserve utility, as $U_1^{FP}(p_h) = U_2^{FP}(p_h) = 0$.

### 3.4. First Score

The nature of the first-score strategy is described as follows: first, PV firms offering high and low prices can be awarded contracts. Second, $Q$ is a variable because quality is deemed heterogeneous, where quality and price matter. Third, quality manipulation occurs. The corruption-proof measures are described in Section 3.4.1. Fourth, the PV firm is incentivized to be corrupt when it is of low quality or unsure if it is a low-price firm. The timeline of the first-score procurement auction is shown in Figure 3, which differs from Figure 2 for items 1, 3, and 4. The auctioneer proposes a first-score bidding rule, and PV firms approach the auctioneer. Auctioneers conduct quality manipulation under external oversight. Furthermore, PV firms submit quality and price bids. The winner benefits from corruption and transfers a part to the auctioneer as a trade-off.

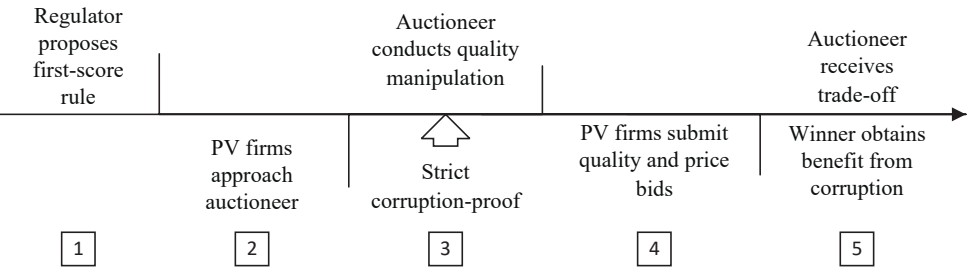

**Figure 3.** Timeline of first-score.

3.4.1. Quality Manipulation

Auctioneer conducts quality manipulation. We assume that the auctioneer is in one of the following states: $s_1(Q_1^{s_1} = Q_h, Q_2^{s_1} = Q_l)$, $s_2(Q_1^{s_2} = Q_l, Q_2^{s_2} = Q_h)$, or $s_0(Q_1^{s_0} = Q_2^{s_0} = \frac{Q_h + Q_l}{2})$, where $Q_h$ and $Q_l$ are the upper and lower boundaries of quality, respectively, $\Delta Q \triangleq Q_h - Q_l$. The auctioneer is honest in $s_1$ or $s_2$ and has income $R_{s_1}$ or $R_{s_2}$. State $s_0$ denotes that the auctioneer has manipulated the quality of the favored PV firm, receives 0 as the retained payoff, and suffers $R_{s_1}$ or $R_{s_2}$ as the loss. For instance, the auctioneer favors Bidder 1, who holds the lower quality. The auctioneer revises both bidders' qualities to $\frac{Q_h + Q_l}{2}$. Thus, Bidder 1 has the same quality level as Bidder 2.

We elaborate a corruption-proof mechanism. The profit that the firm obtains through quality manipulation is $\frac{1}{1+\lambda_f}[U_2^{s_0}(p_l) - U_2^{s_1}(p_l)]$ or $\frac{1}{1+\lambda_f}[U_1^{s_0}(p_l) - U_1^{s_2}(p_l)]$, where $\lambda_f$ represents the cost of secretly transferring payments to the auctioneer. This is exactly the maximum amount of transfer by which the PV producer can be corrupt. The coalition does not exist if the auctioneer's loss $R_{s_1}$ or $R_{s_2}$ exceeds the PV firm's earnings. This is the corruption-proof measure applied to the first-score procurement auction.

3.4.2. Scoring Rule

To be in accordance with the first-price scenario, we define $S(Q, P) - \frac{\alpha}{1-\lambda}(R_{s_1} + R_{s_2})$ as the scoring rule,

$$
\begin{aligned}
S(Q, P) =& \mathbb{E}_{P_i}\left[x_1(P_1, P_2)\left(Q_1 - \frac{1}{1-\lambda}(P_1 - c_1(P_1, P_2) + \psi(c_1(P_1, P_2)))\right)\right] - \frac{1}{1-\lambda}\mathbb{E}_{p_j}[U_1(p_j)] \\
&+ \mathbb{E}_{P_i}\left[x_2(P_1, P_2)\left(Q_2 - \frac{1}{1-\lambda}(P_2 - c_2(P_1, P_2) + \psi(c_2(P_1, P_2)))\right)\right] - \frac{1}{1-\lambda}\mathbb{E}_{p_j}[U_2(p_j)] \\
&- \frac{R}{1-\lambda}.
\end{aligned}
\tag{4}
$$

where $\frac{\alpha}{1-\lambda}(R_{s_1} + R_{s_2})$ is a transfer to the auctioneer when it does not engage in quality manipulation, $j = h, l$. Equation (5a) optimizes the benefits of users, while IC$_1$ and IC$_2$ maximize firms' utilities and IR$_1$ ensures their participation. Equation (5b) present IC$_3$ and IC$_4$ constraints, respectively, to resist corruption coalitions.

$$\max_{x_i^{FS}(\cdot,\cdot),c_i^{FS}(\cdot,\cdot)} S(Q,P) - \frac{\alpha}{1-\lambda}(R_{s_1} + R_{s_2}) \tag{5a}$$

$$s.t. \begin{cases} U_1(p_l) = \mathbb{E}_{P_2}[x_1(p_h, P_2)\Phi(c_1(p_h, P_2))] : (IC_1) \\ U_2(p_l) = \mathbb{E}_{P_1}[x_2(P_1, p_h)\Phi(c_2(P_1, p_h))] : (IC_2) \\ U_i(p_j) \geqslant 0 : (IR_1), \ i = 1, 2; \ j = h, l \\ R_{s_1} \geqslant \frac{1}{1+\lambda_f}\left[U_2^{s_0}(p_l) - U_2^{s_1}(p_l)\right] : (IC_3) \\ R_{s_2} \geqslant \frac{1}{1+\lambda_f}\left[U_1^{s_0}(p_l) - U_1^{s_2}(p_l)\right] : (IC_4) \\ \sum_{i=1}^{2} x_i(P_1, P_2) = 1, x_i(P_1, P_2) \geqslant 0, \end{cases} \tag{5b}$$

where $S(Q,P) = \delta S_{s_1}(Q,P) + \delta S_{s_2}(Q,P) + (1-\delta)S_{s_0}(Q,P)$ represents the expectations of $S(Q,P)$ for all three information states.

Proposition 2 presents the features of the benefits from corruption under the first-score condition.

**Proposition 2.** *In the first-score case, the benefit from the corruption of the winning firm is fixed to the highest level of $c^*$ when it is a low-price firm; the benefits of a high-price firm under $s_1$ and $s_2$ drop to $\tilde{c}$ and $\hat{c}$; a high-price firm in status $s_0$ obtains a benefit $\check{c}$. $c^*$, $\tilde{c}$, $\hat{c}$, and $\check{c}$ are the solutions of $\psi'(c) = 1$, $\psi'(\tilde{c}) = 1 - \frac{\alpha}{1-\alpha}\Phi'(\tilde{c})$, $\psi'(\hat{c}) = 1 - \frac{\alpha\lambda_f}{(1-\alpha)(1+\lambda_f)}\Phi'(\hat{c})$ and $\psi'(\check{c}) = 1 - \frac{\alpha(2+\lambda_f)}{(1+\lambda_f)(1-\alpha)}\Phi'(\hat{c})$, respectively.*

It is worth noting that both high- and low-price firms can win first-score PV procurement auctions. This is the main difference from the first-price scenario. We observe that a low-price winner maintains the highest benefit from corruption. The high-price bidder's benefit from corruption is lower than that of the low-price bidder. Appendix B.2 attaches the proof of Proposition 2.

## 4. Equilibrium Analysis on Model's Outcome

We employ the following data for equilibrium analyses. China has announced the abolition of price subsidies for PV power plants in 2021 [50]. Therefore, this study assumes the prices for desulphurized coal as the benchmark price, which are equal to 53 (USD/MWh) and 65 (USD/MWh) for the lower and upper boundaries of PV firms' prices, respectively. (The price range from 0.37 to 0.45 CNY/KWh (tax included) for newly installed PV plants is approximately equal to 53 to 65 USD/MWh at the current exchange rate of 698.97 [51]). According to the National Energy Administration's PV policy implemented since 2018, (See the "Notice of the National Energy Administration, the Ministry of Finance, and the National Development and Reform Commission on matters related to PV power generation in 2018" [52]) there are three quality benchmarks for different areas, ranging from 70 to 100 (USD/MWh). (The benchmark quality ranging from 0.5 to 0.7 CNY/KWh (tax included) for newly installed PV power plants is approximately equal to 70 to 100 USD/MWh at the current exchange rate of 698.97 [51].) Therefore, users can receive a maximum of $Q_h = 100$(USD/MWh) and a minimum of $Q_l = 70$(USD/MWh) by using PV power. Additionally, we use the observed average tax rate of a typical power plant (See the "Report of Huaneng Power International Co., Ltd., 2022Q3" [53]). to reflect the parameter $\lambda$, which is set to 3%, and the cost of corruption transfer as $\lambda_f = 2\%$ because the annual cost of corruption is 2% of the global GDP [54]. Moreover, we assume that the auctioneer has a prior probability of $\alpha = 0.5$ that a PV firm is low-price and that the rate of disutility of corruption

is $k = 0.8$. All three information states have a probability of $\delta = \frac{1}{3}$. The parameter values used for equilibrium analysis are summarized in Table 2. We simulate the equilibrium solutions for the first-price and first-score scenarios using MATLAB. The simulation adopts $\psi(c) = ke^c$ as the disutility function, which satisfies $\psi' \geqslant 0$, $\psi'' \geqslant 0$ and will deduce similar results as $\psi(c) = kc^2$, $\psi(c) = kc^3$ or the other function types.

**Table 2.** Data for equilibrium analyses.

| Parameters | $\lambda$ | $\lambda_f$ | $\alpha$ | $k$ | $\delta$ | $p_h$ | $p_l$ | $Q_h$ | $Q_l$ |
|---|---|---|---|---|---|---|---|---|---|
| | | | | | | (USD/MWh) | | | |
| Values | 3% | 2% | 0.5 | 0.8 | $\frac{1}{3}$ | 65 | 53 | 100 | 70 |

### 4.1. Social Welfare

Given the strict power quality requirements for PV projects, the equilibrium analyses employ qualities $Q_h$ and $Q_l$ as two dimensions for the study. Additionally, $p_h$ and $p_l$ are better fitting parameters for the equilibrium analysis because many regulators adopt first-score procurement auctions and do not currently exhibit high prices.

The highest level of social welfare is achieved when the regulator selects the first-score scenario. Figure 4a shows that social welfare with quality. Figure 4a illustrates the relationship between social welfare $S$ and the upper and lower boundaries, $Q_h$ and $Q_l$, of PV enterprise quality. Figure 4b illustrates the relationship between social welfare $S$ and the upper and lower boundaries, $p_h$ and $p_l$, of PV enterprise price. Figure 4a shows that the first-price concept is not optimal under any parameter condition when the regulator of the PV scheme has a strong preference for quality. This result holds true when the regulator applies strict corruption-proof measures in both the first-price and first-score scenarios. Moreover, Figure 4a shows that the first-price strategy leads to increased social welfare in low-quality cases. In contrast, the results also suggest that the first-score approach can be a rational choice for the regulator, as PV power generation continues to increase in quality over time.

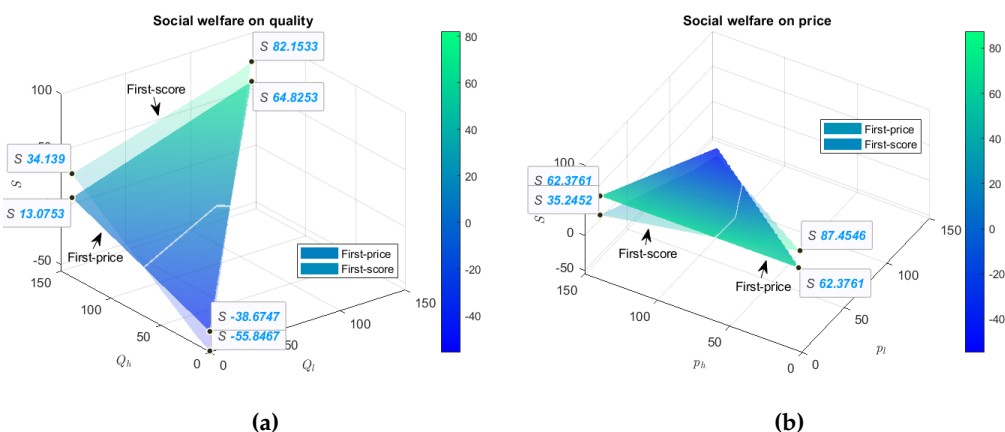

**Figure 4.** Social welfare on quality and price (**a**) social welfare on quality, (**b**) social welfare on price.

Figure 4b reveals that the first-price concept performs well when the price of PV power generation increases, especially when $p > 100$ (USD/MWh). Therefore, the regulator is not incentivized to ensure the first-price strategy for a wide variety of parameter settings despite its favorability when the price soars.

### 4.2. Corruption Benefit of PV Firm

As stated in Proposition 2 and illustrated in Figure 5, the first-price approach is more likely to achieve a higher benefit from corruption than the first-score scenario.

Figure 5a illustrates the relationship between corruption benefit $c$ and the upper and lower boundaries, $Q_h$ and $Q_l$, of PV enterprise quality. Figure 5b illustrates the relationship between corruption benefit $c$ and the upper and lower boundaries, $p_h$ and $p_l$, of PV enterprise price. Analyses of corruption reveal that the first-score strategy results in PV firms' negative benefits, that is, punishment. This is due to the application of strict corruption-proof measures on quality manipulation. When PV products' quality competition arises, that is, $Q_h \approx Q_l$, the first-score strategy causes the PV firm to suffer less punishment because quality manipulation is easy to conduct. However, the price competition, that is, $p_h \approx p_l$, of the first-score scenario lowers PV firms' benefit from corruption.

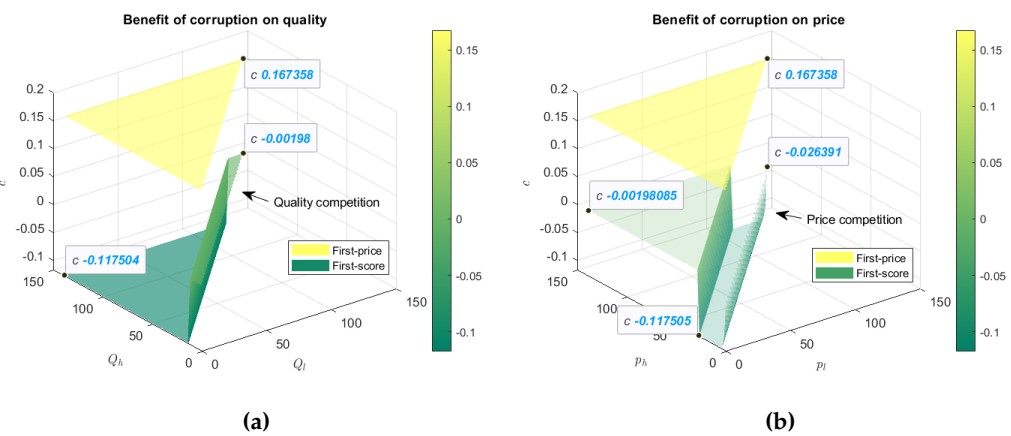

**Figure 5.** Benefit of corruption on quality and price (**a**) benefit of corruption on quality, (**b**) benefit of corruption on price.

### 4.3. Utility of PV Firm

The expected utility of the PV firm in the first-score scenario is greater than that for the first-price case. The contractor has the least utility in the first-price situation relative to that of the first-score.

As stated in Proposition 1 and shown in Figure 6, the utility of the PV firm is normalized to a lower level in the first-price scenario. Figure 6a illustrates the relationship between utility $U$ of PV firm and the upper and lower boundaries, $Q_h$ and $Q_l$, of PV enterprise quality. Figure 6b illustrates the relationship between utility $U$ of PV firm and the upper and lower boundaries, $p_h$ and $p_l$, of PV enterprise price. It is desirable for the regulator to employ the first-score approach, even though a distortion occurs when there are quality and price competitions, that is, $Q_h \approx Q_l$ and $p_h \approx p_l$. Quality(price) competition decreases(increases) a PV firm's utility. This is because $U_i(p_j) = t_i - x_i(P_1, P_2)\psi(c_i(P_1, P_2))$ is negatively related with $c_i(P_1, P_2)$, which decreases, as Figure 5b shows. This result remains true, regardless of the quality and price of the scenarios involved.

### 4.4. Total Income of PV Firm

As shown in Figure 7, the first-price scenario obtains a greater income than that of the first-score. Figure 7a illustrates the relationship between total income $U + c$ of PV firm and the upper and lower boundaries, $Q_h$ and $Q_l$, of PV enterprise quality. Figure 7b illustrates the relationship between total income $U + c$ of PV firm and the upper and lower boundaries, $p_h$ and $p_l$, of PV enterprise price. A PV firm's total income comprises its utility and the benefits of corruption. Therefore, the first-price approach is a better choice for PV firms in both quality and price settings. The first-score strategy is inadvisable for a PV firm to increase its total income. These results are rational for both quality and price settings.

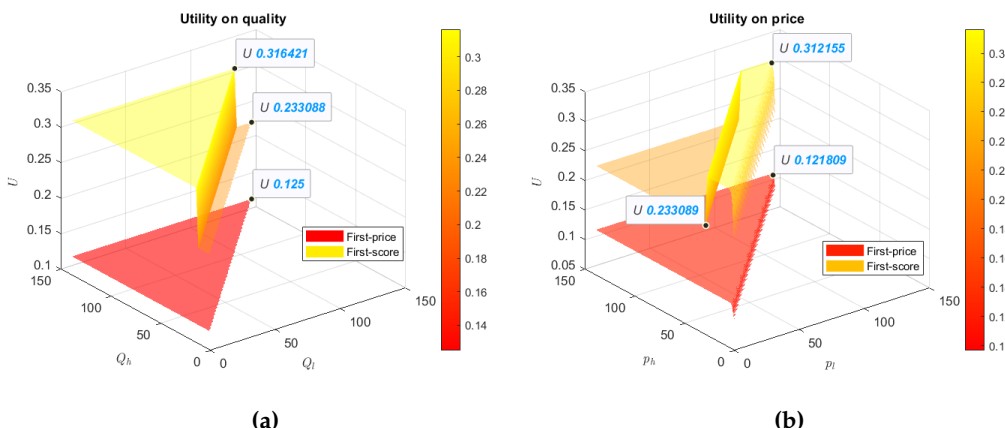

**Figure 6.** Utility of PV firm on quality and price (**a**) utility of PV firm on quality, (**b**) utility of PV firm on price.

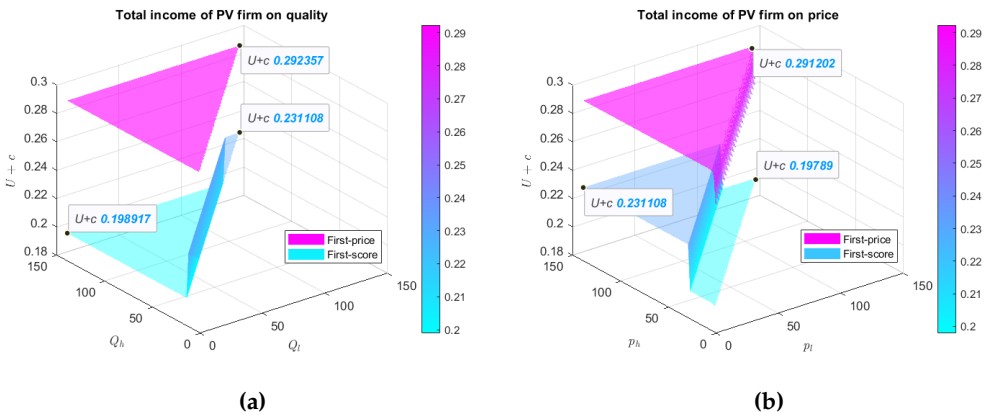

**Figure 7.** Total income of PV firm on quality and price (**a**) total income of PV firm on quality, (**b**) total income of PV firm on price.

*4.5. Impact of $\lambda$ and $\lambda_f$*

As shown in Figure 8, the first-score policy achieves high performance in social welfare. However, the first-price policy is good for PV firms to benefit from corruption. Figure 8 illustrates the impact of tax rate $\lambda$ and corruption cost $\lambda_f$ on social welfare $S$ (Figure 8a), utility $U$ of PV firm (Figure 8b), corruption benefit $c$ (Figure 8c) and total income $U + c$ of PV firm (Figure 8d). Figure 8a shows that the first-price option is no longer the best for social welfare. The first-score alternative is always the best for improving utility (Figure 8b). Moreover, the winner of the first-price scenario obtains the highest benefit from corruption (Figure 8c) and total income (Figure 8d) of a PV firm.

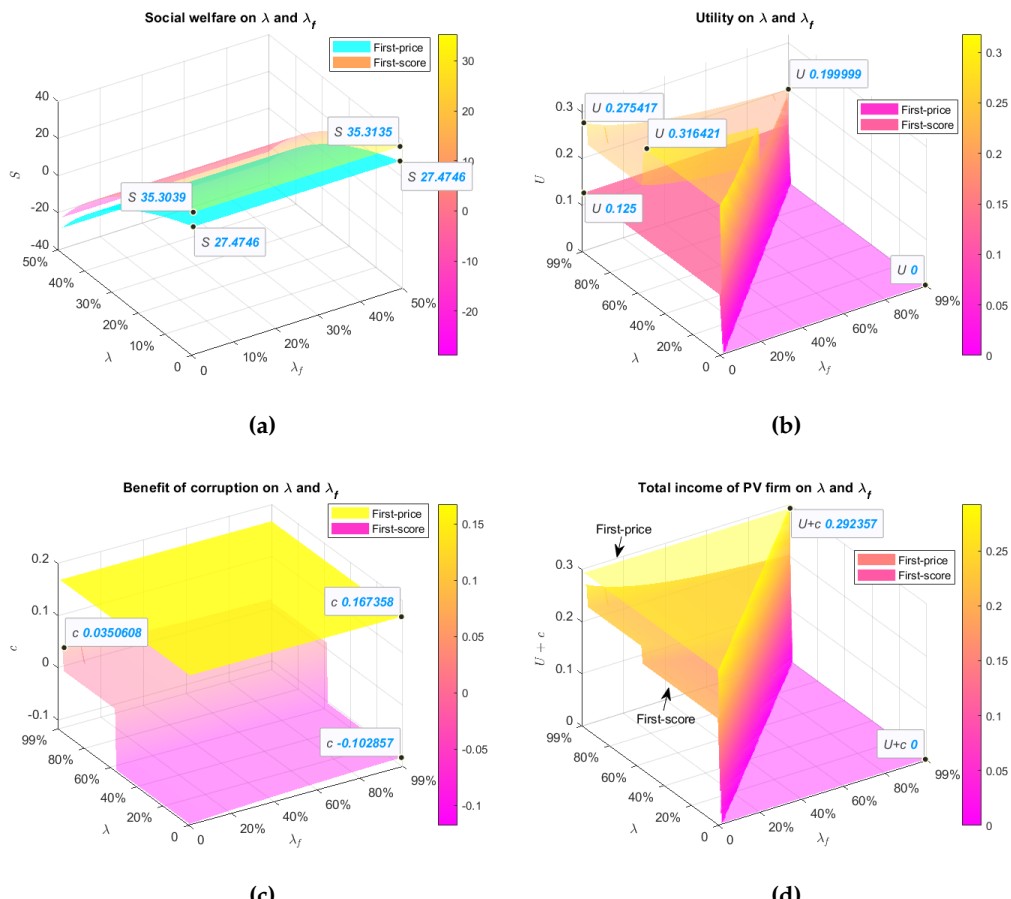

**Figure 8.** Impact of $\lambda$ and $\lambda_f$ (**a**) impact of $\lambda$ and $\lambda_f$ on social welfare, (**b**) impact of $\lambda$ and $\lambda_f$ on utility, (**c**) impact of $\lambda$ and $\lambda_f$ on benefit of corruption, (**d**) impact of $\lambda$ and $\lambda_f$ on total income of PV firm.

### 4.6. Impact of k and α

Social welfare shows a positive relationship with the possibility of a low price, *a*, and appears to be almost unchanged based on the rate of disutility of corruption, *k*, as shown in Figure 9a. Figure 9 illustrates the impact of the rate *k* of corruption's disutility and the probability *α* of low price on social welfare *S* (Figure 9a), utility *U* of PV firm (Figure 9b), corruption benefit *c* (Figure 9c) and total income *U* + *c* of PV firm (Figure 9d). This shows a negative correlation between the social welfare, *S*, and the possibility of a low price, *a*. Additionally, the proceeds of corruption of PV firms increase with *α* in the first-price scenario and remain unchanged in the first-score case, which explains the results shown in Figure 9c. The higher the possibility of a low price, *α*, the more likely it is that the PV firm will engage in corrupt practices with the auctioneer. In addition, we find that the utility of a PV firm negatively correlates with *k* and *α*, as Figure 9b illustrates. This finding also applies to the analysis of the total revenue of PV enterprises (Figure 9d).

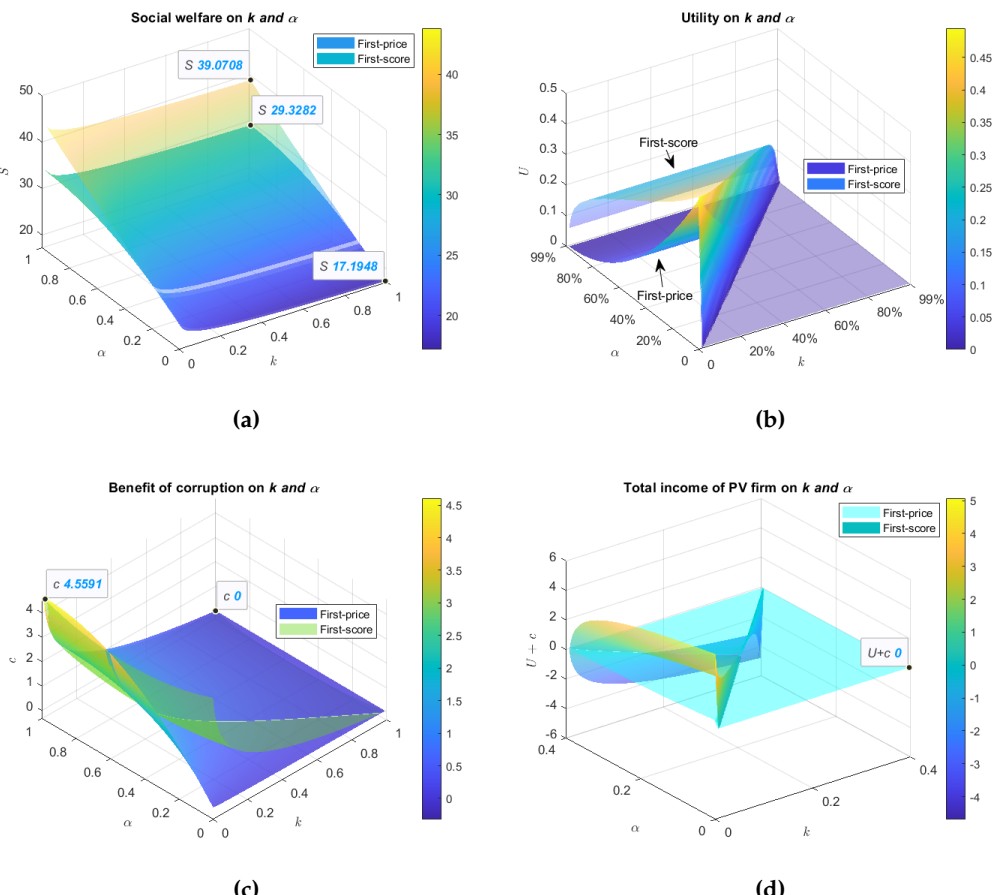

**Figure 9.** Impact of *k* and *α* (**a**) impact of *k* and *α* on social welfare, (**b**) impact of *k* and *α* on utility, (**c**) impact of *k* and *α* on benefit of corruption, (**d**) impact of *k* and *α* on total income of PV firm.

### 4.7. Impact of k and δ

Figure 10a explains the positive change in social welfare with $\delta$. Figure 10 illustrates the impact of the rate *k* of corruption's dis-utility and the probability $\delta$ of information states on social welfare *S* (Figure 10a), utility *U* of PV firm (Figure 10b), corruption benefit *c* (Figure 10c) and total income $U + c$ of PV firm (Figure 10d). $\delta$ denotes the degree of corruption. The higher the $\delta$, the lower the incentive for PV companies to engage in corruption. This once again proves that, in the environment of preventing corruption, the first-price policy is inappropriate for PV companies. Furthermore, PV firms should be honest in order to realize the utmost utility ( Figure 10b). Corrupt bidding practices allow PV companies to reap more corrupt benefits than honest practices, as Figure 10c shows. The highest benefit of corruption achieved in corrupt practices makes PV firms receive the utmost total income (Figure 10d).

In summary, the equilibrium analyses show that all cases share the same results. The first-score strategy is suitable for high-quality and low-price PV markets because it can achieve the utmost social welfare in most settings. This is also beneficial to the utility and total revenue of PV enterprises. The first-price approach and corrupt bidding practice of the first-score strategy favor the proceeds of corruption. This finding applies to most settings of tax rate, cost of interal transfer, low price likelihood, corruption disutility, and corruption degree.

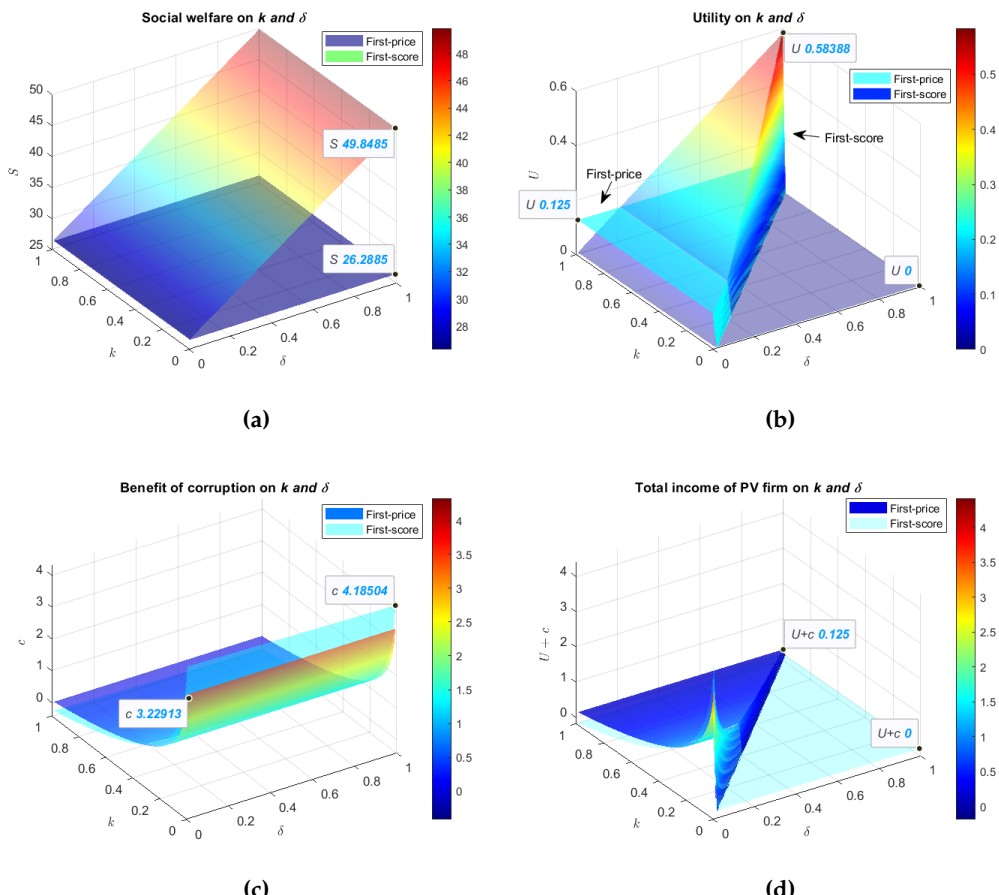

**Figure 10.** Impact of $k$ and $\delta$ (**a**) impact of $k$ and $\delta$ on social welfare, (**b**) impact of $k$ and $\delta$ on utility, (**c**) impact of $k$ and $\delta$ on benefit of corruption, (**d**) impact of $k$ and $\delta$ on total income of PV firm.

## 5. Discussion

Corruption has become an important obstacle hindering the development of PV power generation incentive schemes. This study employs a mechanism design approach and a Bayesian equilibrium model to explore this issue. We also investigate ways to avoid corruption by pursuing refined solutions.

We consider a strict corruption-proof PV procurement auction environment. We develop two PV procurement auction mechanisms to investigate which scenario promotes social welfare. One is the first-price strategy, and the other is the first-score approach. We find that the first-score scenario can sustain social welfare at a high level in most settings. Under this framework, we investigate the scenario that maximizes PV firms' utility and total income. We find that the first-price policy is inappropriate for PV enterprises because it inhibits their normal utility of PV enterprises. Furthermore, we study the scenario that maximizes the PV firm's corruption benefit. We find that the first-price policy is inadvisable because it causes the PV firm to receive the most corrupt revenue.

The theoretical and simulation analyses result in the following suggestions, which have guiding significance for supervision measures. First, the first-price approach entails outstanding social welfare in a low-quality and high-price PV market. However, because the first-price strategy enables the contractor to maintain the utmost benefit from corruption, a regulator may not employ it. These are the advantages and disadvantages of the first-price concept, which answer Q1. Second, in most market environments, the first-score strategy can obtain more prominent social benefits than that of the first-price. However, the first-score approach is also conducive to PV enterprises obtaining maximum utility.

These two implications answer Q2 that regulators should employ a first-score mechanism to maximize social welfare, especially in a high-quality and low-price PV market.

Comparing our results with those of previous studies leads to the following findings. First, the  most closely literature related to our study is Wang [12], which compares first-score and second-score auctions. In our study we find that in the case of first-score, when the winning firm is a low-price firm, its income from corruption is fixed at the highest level. In this regards Wang [12] argues that more efficient suppliers are willing to pay a higher bribe in first-score auction. In contrast, our study differs from Wang [12] since we deal with first-price auction and find that a higher benefit from corruption is likely to occur than the first-score scenario. Wang [12] uncovers that the second-score auction leads to a higher equilibrium bribe and thus is more vulnerable to corruption. Second, the works of Burguet [29] and Huang and Xia [32] are also relevant to further compare and strengthen our findings. In his insightful study, Burguet [29] performs an analytical deduction revealing that optimal contracts limit quality to all contractors. To resolve the trade-off between corruption deterrence and quality distortion, Huang and Xia [32] find that the buyer may overstate his/her preference for quality. Our study differs from the above authors since we consider additional indicators as proxy of social welfare, utility, corruption income and total revenue of PV enterprises. In conclusion, our results are consistent with those of auction comparison and illustrate some novel insights.

This study has some contributions in terms of empirical research results. We highlight the contributions as follows. First, the first-price approach entails outstanding social welfare in a low-quality and high-price PV market. Second, regulators should be very cautious about using this strategy, as the first-price strategy favors corruption and enables contractors to maintain maximum benefits from corruption. Third, the first-score strategy is suitable for the high-quality and low-price PV market, because it can achieve the highest social welfare in most cases. This is also beneficial to the utility and total revenue of PV enterprises. For this reason, first-score strategy is suitable for high-quality and low-price PV markets. Forth, first-score approach is conducive to PV enterprises obtaining maximum utility. As a result, regulators should employ a first-score mechanism to maximize social welfare, especially in a high-quality and low-price PV market.

This study contributes to future research in the context of anti-corruption and the impact of PV power generation. First of all, this study adopts a first-score auction strategy to study the common quality manipulation problems in the field of PV procurement auction. Future studies can use this method to study issues related to PV power generation. Problems include but are not limited to wind farms and energy storage projects. Secondly, our cost remuneration assumption is applicable to anti-corruption research in procurement regulation. Future studies can use these assumptions to reflect the relationship between PV enterprises and users. Third, our mechanism is designed to screen out asymmetric information about bid prices. In this way, regulators can get bidders to tell their truth and get the most out of them. Future anti-corruption research can use this mechanism as a baseline of anti-corruption model framework.

This study can be applied to procurement auctions in other renewable energy projects. These latter may include for example procurement auctions of wind power station or solar energy power station. It also applies to auctions of new energy sources, such as liquefied natural gas storage power station. However, this study does not apply to traditional energy sources, such as thermal power, whose procurement mode is not purchasing but dispatching. Given the requirement of thermal power withdrawal, new thermal power plant projects will be rare in the future. As a result, purchasing projects for thermal power plants will gradually decrease. Therefore, our study is only applicable for renewable power generation station projects.

## 6. Conclusions

In summary, our study is innovative in several ways. Firstly, we arrange the content according to the findings of literature research. Price manipulation is hard to sustain in

tightly regulated auctions for PV purchases. This makes it unnecessary to study price manipulation in both first-price and first-score PV procurement auctions. However, we conduct our study on quality manipulation because it is prevalent and well-distributed among recent corrupt studies in first-score auctions. We also find that research on power generation rarely deals with corruption in PV procurement auctions. Therefore, we establish a framework using the principle of revelation and propose measures to prevent corruption in PV procurement auctions. Besides, policies to curb PV corruption require further research to adapt to different markets. To this end, we have developed two auction markets, high-quality-low-price and low-quality-high-price, and formulated policies to curb PV corruption.

Secondly, by comparing the first-price auction with the first-score auction, we find the appropriate strategy to restrain the corruption of PV procurement auction. We used a first-score scheme as research object. The reason is two-fold. First, in recent years, there is a widespread quality manipulation in the auction industry. The second is the large-scale application of the first-score auction in the PV procurement auction. In order to reveal the nature of the first-score auction, this paper establishes scoring rules that reflect the regulator's maximum expectations for quality and price. Next, we define three states of information that reveal the auctioneer's unethical behavior in terms of quality manipulation. To reduce corruption, an IC constraint was added to the first-score model to ensure that the auctioneer had no incentive to report falsified quality assessments. In equilibrium analysis, first-price and first-score of PV procurement auctions are compared in social welfare, utility, corruption benefit and total income.

Finally, we use Bayesian Nash equilibrium, revelation principle and exogenous favoritism to arrange the study. The motivation is threefold. First, Nash equilibrium solves the game problem by deriving an equilibrium between the strategies of two players. We define the objective function of the user and the PV enterprise, expressed in terms of social welfare and optimal utility and we use an implicit variable $U$ to link the two. Secondly, the revelation principle is used to solve the Bayesian Nash equilibrium. By using the IC constraint, PV enterprise is of true type (i.e., it is not corrupted) and obtains the best utility. In addition, we have added IR constraints to ensure the participation of PV companies. Thirdly, we use exogenous favoritism to reflect the reality of procurement auctions. We describe a situation in which the auctioneer does not favor any company in the bidding process; the winning bidder leaves part of the corruption process to the auctioneer as remuneration; the auctioneer could not distinguish the winner beforehand. Therefore, the application of exogenous favoritism reflects the reality of auction practice.

This study has certain limitations. Future work should focus on a more dynamic study because recursive analysis would be more suitable to describe the long-term effects of a tendering scheme and regulatory policy for the PV market. Additionally, an empirical analysis would be beneficial to strengthen the findings of a numerical and equilibrium computations. Furthermore, additional insight can likely be obtained through the evaluation of hidden behavior cases, such as the trade-off among bidders and auctioneers. This can result in a model that can better reflect existing bidding programs.

**Author Contributions:** Conceptualization, P.H. and J.-P.G.; methodology, P.H.; software, P.H.; validation, P.H., J.-P.G. and E.O.; formal analysis, P.H.; investigation, P.H.; resources, Y.-H.S.; data curation, P.H.; writing—original draft preparation, P.H.; writing—review and editing, E.O.; visualization, Y.-H.S.; supervision, J.-P.G.; project administration, J.-P.G.; funding acquisition, P.H. All authors have read and agreed to the published version of the manuscript.

**Funding:** This research was funded by the General Project of the Humanities and Social Science Fund of the Chinese Ministry of Education "A study on optimization mechanism and policy coordination of power system grid source towards carbon neutral goal" under Grant No. 21YJA630023; the Major Project of the National Social Science Fund "Research on the path and collaborative mechanism of energy structure transformation under the 'two-carbon' target" under Grant No. 22&ZD104; Special Project of Policy Modeling and Strategy Research supported by National Dual-Carbon Strategy Supported by National Natural Science Foundation of China Key project "Research on collaborative

development optimization mechanism, dispatching strategy and policy of Smart Power Grid oriented to carbon neutrality" under Grant No. 72243009; National Natural Science Foundation of China under Grant No. 72171165, 72174141, and 71874121.

**Institutional Review Board Statement:** Not applicable.

**Informed Consent Statement:** Not applicable.

**Data Availability Statement:** All data employed in this paper are provided in the Section of "Equilibrium analysis".

**Acknowledgments:** The authors acknowledge the administrative and technical support from Tianjin Renai College and Tianjin University.

**Conflicts of Interest:** The authors declare no conflict of interest.

## Abbreviations

The following abbreviations are used in this manuscript:

| | |
|---|---|
| FP | First-price |
| FS | First-score |
| IC | Incentive compatibility |
| IR | Individual rationality |

## Appendix A

*Appendix A.1. Proof of Lemma 1*

**Proof.** IC constraint of $U_i(p_l)$ leads to

$$
\begin{aligned}
U_1(p_l) = \quad & \mathbb{E}_{P_2}[t_1(p_l, P_2) - x_1(p_l, P_2)\psi(p_l - C_1(p_l, P_2))] \geqslant \\
& \mathbb{E}_{P_2}[t_1(p_h, P_2) - x_1(p_h, P_2)\psi(p_l - C_1(p_h, P_2))],
\end{aligned} \tag{A1}
$$

when the price of Bidder 1 is $p_l$, and

$$
\begin{aligned}
U_2(p_l) = \quad & \mathbb{E}_{P_1}[t_2(P_1, p_l) - x_2(P_1, p_l)\psi(p_l - C_2(P_1, p_l))] \geqslant \\
& \mathbb{E}_{P_1}[t_2(P_1, p_h) - x_2(P_1, p_h)\psi(p_l - C_2(P_1, p_h))],
\end{aligned} \tag{A2}
$$

when the price of Bidder 2 is $p_l$. When the type of Bidders 1 or 2 is $p_h$, the IR constraint requires the expected utility for each firm to be equal to or larger than 0, that is,

$$
U_1(p_h) = \mathbb{E}_{P_2}[t_1(p_h, P_2) - x_1(p_h, P_2)\psi(p_h - C_1(p_h, P_2))] \geqslant 0, \tag{A3a}
$$
$$
U_2(p_h) = \mathbb{E}_{P_1}[t_2(P_1, p_h) - x_2(P_1, p_h)\psi(p_h - C_2(P_1, p_h))] \geqslant 0. \tag{A3b}
$$

Next, we prove in 1° that the high-price firm obtains only reservation utility, that is, $U_i(p_h) = 0$, and derive an alternative IC constraint of $U_i(p_l)$ in 2°.

1° The maximization of user surplus (2) requires that $U_i(p_j)$ takes the value at its lower boundary. Consequently, it is binding for (A1), (A2), (A3a), and (A3b). (A3a) and (A3b) indicate that a firm with a high price only receives reservation utility, that is,

$$
U_1(p_h) = 0, \quad U_2(p_h) = 0. \tag{A4}
$$

$2°$ $U_1(p_l)$ (A1) can be transformed into

$$U_1(p_l) = \mathbb{E}_{P_2}\left[\underbrace{t_1(p_h, P_2) - x_1(p_h, P_2)\psi(p_l - C_1(p_h, P_2))}_{\text{the right hand side of (A1)}}\right] \tag{A5}$$

$$= \mathbb{E}_{P_2}\left[\underbrace{x_1(p_h, P_2)\psi(p_h - C_1(p_h, P_2))}_{\text{the left hand side of (A3a)}} - x_1(p_h, P_2)\psi(p_l - C_1(p_h, P_2))\right]$$

$$= \mathbb{E}_{P_2}[x_1(p_h, P_2)(\psi(p_h - C_1(p_h, P_2)) - \psi(p_l - C_1(p_h, P_2)))]$$

$$= \mathbb{E}_{P_2}[x_1(p_h, P_2)(\psi(c_1(p_h, P_2)) - \psi(c_1(p_h, P_2) - \Delta p))]$$

$$= \mathbb{E}_{P_2}[x_1(p_h, P_2)\Phi(c_1(p_h, P_2))],$$

where $\Phi(c) = \psi(c) - \psi(c - \Delta p)$, $\Phi(c_1(p_h, P_2)) = \psi(c_1(p_h, P_2)) - \psi(c_1(p_h, P_2) - \Delta p)$. Similarly, $U_2(p_l)$ has expectation of $\mathbb{E}_{P_1}[x_2(P_1, p_h)\Phi(c_2(P_1, p_h))]$. □

## Appendix B

*Appendix B.1. Proof of Proposition 1*

**Proof.** The objective function of the first-price scenario presented in (A6) is to maximize $S^{FP}$ with respect to $x_i^{FP}(\cdot), c_i^{FP}(\cdot)$.

$$\max_{x_i^{FP}(\cdot), c_i^{FP}(\cdot)} S^{FP} = Q - \frac{1}{1-\lambda}\mathbb{E}_{P_i}\left[x_1(P_1, P_2)(P_1 - c_1(P_1, P_2) + \psi(c_1(P_1, P_2))) + \mathbb{E}_{p_j}[U_1(p_j)]\right] \tag{A6}$$

$$- \frac{1}{1-\lambda}\mathbb{E}_{P_i}\left[x_2(P_1, P_2)(P_2 - c_2(P_1, P_2) + \psi(c_2(P_1, P_2))) + \mathbb{E}_{p_j}[U_2(p_j)]\right]$$

$$- \frac{R}{1-\lambda},$$

where $\mathbb{E}_{p_j}[U_1(p_j)] = \alpha U_1(p_l)$ and $\mathbb{E}_{p_j}[U_2(p_j)] = \alpha U_2(p_l)$. As the constant term $-\frac{R}{1-\lambda}$ has no effect on the derivative term, ignoring the constant term has no impact on the result. The expected user's surplus $S^{FP}$ can be represented as (A7) based on $U_1(p_l) = \alpha x_1(p_h, p_l)\Phi(c_1(p_h, p_l)) + (1 - \alpha)x_1(p_h, p_h)\Phi(c_1(p_h, p_h))$ and $U_2(p_l) = \alpha x_2(p_l, p_h)\Phi(c_2(p_l, p_h)) + (1 - \alpha)x_2(p_h, p_h)\Phi(c_2(p_h, p_h))$.

$$\max\left\{\alpha^2\left[Q - (\frac{1}{1-\lambda})[p_l - c_1(p_l, p_l) + \psi(c_1(p_l, p_l))]\right]x_1(p_l, p_l)\right. \tag{A7}$$

$$+ \alpha^2\left[Q - (\frac{1}{1-\lambda})[p_l - c_2(p_l, p_l) + \psi(c_2(p_l, p_l))]\right]x_2(p_l, p_l)$$

$$+ \alpha(1-\alpha)\left[Q - (\frac{1}{1-\lambda})[p_l - c_1(p_l, p_h) + \psi(c_1(p_l, p_h))]\right]x_1(p_l, p_h)$$

$$+ \alpha(1-\alpha)\left[Q - (\frac{1}{1-\lambda})\left[p_h - c_2(p_l, p_h) + \psi(c_2(p_l, p_h)) + \frac{\alpha}{1-\alpha}\Phi(c_2(p_l, p_h))\right]\right]x_2(p_l, p_h)$$

$$+ \alpha(1-\alpha)\left[Q - (\frac{1}{1-\lambda})\left[p_h - c_1(p_h, p_l) + \psi(c_1(p_h, p_l)) + \frac{\alpha}{1-\alpha}\Phi(c_1(p_h, p_l))\right]\right]x_1(p_h, p_l)$$

$$+ \alpha(1-\alpha)\left[Q - (\frac{1}{1-\lambda})[p_l - c_2(p_h, p_l) + \psi(c_2(p_h, p_l))]\right]x_2(p_h, p_l)$$

$$+ (1-\alpha)^2\left[Q - (\frac{1}{1-\lambda})\left[p_h - c_1(p_h, p_h) + \psi(c_1(p_h, p_h)) + \frac{\alpha}{1-\alpha}\Phi(c_1(p_h, p_h))\right]\right]x_1(p_h, p_h)$$

$$+ (1-\alpha)^2\left[Q - (\frac{1}{1-\lambda})\left[p_h - c_2(p_h, p_h) + \psi(c_2(p_h, p_h)) + \frac{\alpha}{1-\alpha}\Phi(c_2(p_h, p_h))\right]\right]x_2(p_h, p_h)\left.\right\}.$$

The maximization requires that every part in the square brackets be maximized. From the derivation of the first term, we obtain $c_1(p_l, p_l) = c_2(p_l, p_l) = c^*$, where $\psi'(c^*) = 1$, because it has a second-order derivative, $-\frac{1}{1-\lambda}\psi'' < 0$. Analyses of the third and fourth terms yield $c_1(p_l, p_h) = c^*, c_2(p_l, p_h) = \tilde{c}$, where $\psi'(\tilde{c}) = 1 - \frac{\alpha}{1-\alpha}\Phi'(\tilde{c})$. There is $\tilde{c} < c^*$ because $\Phi' > 0$. The results of the fifth to eighth terms can be obtained as follows: $c_1(p_h, p_l) = c_1(p_h, p_h) = c_2(p_h, p_h) = \tilde{c}, c_2(p_h, p_l) = c^*$.

There is $x_1(p_l, p_h) = x_2(p_h, p_l) = 1$ because the first-price scenario excludes high-price firms. The first two terms are equivalent to each other. This leads to $x_1(p_l, p_l) = x_2(p_l, p_l) = \frac{1}{2}$. By substituting $c_1(p_h, p_h)$ and $c_2(p_h, p_h)$ with $\tilde{c}$ and $\tilde{c}$, we find that the difference between the square brackets of the seventh and eighth terms is zero. This deduces $x_1(p_h, p_h) = x_2(p_h, p_h) = \frac{1}{2}$. These lead to $U_1(p_l) = \mathbb{E}_{P_2}[x_1(p_h, P_2)\Phi(c_1(p_h, P_2))] = \alpha \underbrace{x_1(p_h, p_l)}_{=0}\Phi(c_1(p_h, p_l)) + (1-\alpha)\underbrace{x_1(p_h, p_h)}_{=\frac{1}{2}}\Phi(c_1(p_h, p_h)) = \frac{1}{2}(1-\alpha)\Phi(c_1(p_h, p_h))$

and $U_2(p_l) = \mathbb{E}_{P_1}[x_2(P_1, p_h)\Phi(c_2(P_1, p_h))] = \alpha \underbrace{x_2(p_l, p_h)}_{=0}\Phi(c_2(p_l, p_h))$

$+ (1-\alpha)\underbrace{x_2(p_h, p_h)}_{=\frac{1}{2}}\Phi(c_2(p_h, p_h)) = \frac{1}{2}(1-\alpha)\Phi(c_2(p_h, p_h))$.

The proof $1°$ of Lemma 1 has shown in (A4) that $U_1(p_h) = 0$ and $U_2(p_h) = 0$. These indicate that the high-price firm receives only the retained utility in the first-price scenario. □

**Table A1.** Solutions of $c_i^{FP}(P_1, P_2)$ at first-price.

| $x_i^{FP}(P_1,P_2)$ \ $c_i^{FP}(P_1,P_2)$ | $c_1^{FP}(p_l,p_l)$ | $c_2^{FP}(p_l,p_l)$ | $c_1^{FP}(p_l,p_h)$ | $c_2^{FP}(p_h,p_l)$ | $c_1^{FP}(p_h,p_h)$ | $c_2^{FP}(p_h,p_h)$ |
|---|---|---|---|---|---|---|
| $x_1^{FP}(p_l,p_l) = \frac{1}{2}$ | $c^*$ | $c^*$ | | | | |
| $x_1^{FP}(p_l,p_h) = 1$ | | | $c^*$ | | | |
| $x_2^{FP}(p_h,p_l) = 1$ | | | | $c^*$ | | |
| $x_1^{FP}(p_h,p_h) = \frac{1}{2}$ | | | | | $\tilde{c}$ | $\tilde{c}$ |

*Appendix B.2. Proof of Proposition 2*

**Proof.**

$$S(Q,P) - \frac{\alpha}{1-\lambda}(R_{s_1} + R_{s_2}) = \mathbb{E}_{P_i}\left[x_1(P_1,P_2)\left(Q_1 - \frac{1}{1-\lambda}(P_1 - c_1(P_1,P_2) + \psi(c_1(P_1,P_2)))\right)\right] \tag{A8}$$

$$- \frac{1}{1-\lambda}\mathbb{E}_{p_j}[U_1(p_j)]$$

$$+ \mathbb{E}_{P_i}\left[x_2(P_1,P_2)\left(Q_2 - \frac{1}{1-\lambda}(P_2 - c_2(P_1,P_2) + \psi(c_2(P_1,P_2)))\right)\right]$$

$$- \frac{1}{1-\lambda}\mathbb{E}_{p_j}[U_2(p_j)] - \frac{R}{1-\lambda} - \frac{\alpha}{1-\lambda}(R_{s_1} + R_{s_2})$$

Equation (A8) is the objective function, where $\mathbb{E}_{p_j}[U_1(p_j)] = \alpha U_1(p_l)$ and $\mathbb{E}_{p_j}[U_2(p_j)] = \alpha U_2(p_l)$, $U_1(p_l) = \alpha x_1(p_h, p_l)\Phi(c_1(p_h, p_l)) + (1-\alpha)x_1(p_h, p_h)\Phi(c_1(p_h, p_h))$ and $U_2(p_l) = \alpha x_2(p_l, p_h)\Phi(c_2(p_l, p_h)) + (1-\alpha)x_2(p_h, p_h)\Phi(c_2(p_h, p_h))$. We substitute $R_{s_1}$ and $R_{s_2}$ of (5a) with $\frac{1}{1+\lambda_f}(U_2^{s_0}(p_l) - U_2^{s_1}(p_l))$ and $\frac{1}{1+\lambda_f}(U_1^{s_0}(p_l) - U_1^{s_2}(p_l))$ because constraints $IC_3$ and $IC_4$ (5b) are binding when the objective function is maximized. We omit the constant term $-\frac{R}{1-\lambda}$ that has no influence on the optimal solution. The optimization expression is decomposed into three independent components, which represent

the optimization of (5a) in the scenarios of $s_1, s_2, s_0$, respectively. We substitute $U_1^{s_0}(p_l)$ and $U_2^{s_0}(p_l)$ with $U_1^{s_0}(p_l) = \alpha x_1^{s_0}(p_h, p_l)\Phi(c_1^{s_0}(p_h, p_l)) + (1-\alpha)x_1^{s_0}(p_h, p_h)\Phi(c_1^{s_0}(p_h, p_h))$ and $U_2^{s_0}(p_l) = \alpha x_2^{s_0}(p_l, p_h)\Phi(c_2^{s_0}(p_l, p_h)) + (1-\alpha)x_2^{s_0}(p_h, p_h)\Phi(c_2^{s_0}(p_h, p_h))$, $U_2^{s_1}(p_l)$ and $U_1^{s_2}(p_l)$ with $U_2^{s_1}(p_l) = \alpha x_2^{s_1}(p_l, p_h)\Phi(c_2^{s_1}(p_l, p_h)) + (1-\alpha)x_2^{s_1}(p_h, p_h)\Phi(c_2^{s_1}(p_h, p_h))$ and $U_1^{s_2}(p_l) = \alpha x_1^{s_2}(p_h, p_l)\Phi(c_1^{s_2}(p_h, p_l)) + (1-\alpha)x_1^{s_2}(p_h, p_h)\Phi(c_1^{s_2}(p_h, p_h))$.

$S(Q, P)$ has three states $S_{s_1}(Q, P)$, $S_{s_2}(Q, P)$, and $S_{s_0}(Q, P)$. The first part of the objective function $S(Q, P) - \frac{\alpha}{1-\lambda}(R_{s_1} + R_{s_2})$ with respect to $s_1$ is expressed as follows:

$$
\max \Bigg\{ \alpha^2 \Big[ Q_h - \frac{1}{1-\lambda}[p_l - c_1^{s_1}(p_l, p_l) + \psi(c_1^{s_1}(p_l, p_l))] \Big] x_1^{s_1}(p_l, p_l) \tag{A9}
$$

$$
+ \alpha^2 \Big[ Q_l - \frac{1}{1-\lambda}[p_l - c_2^{s_1}(p_l, p_l) + \psi(c_2^{s_1}(p_l, p_l))] \Big] x_2^{s_1}(p_l, p_l)
$$

$$
+ \alpha(1-\alpha) \Big[ Q_h - \frac{1}{1-\lambda}[p_h - c_1^{s_1}(p_h, p_l) + \psi(c_1^{s_1}(p_h, p_l)) + \frac{\alpha}{1-\alpha}\Phi(c_1^{s_1}(p_h, p_l))] \Big] x_1^{s_1}(p_h, p_l)
$$

$$
+ \alpha(1-\alpha) \Big[ Q_l - \frac{1}{1-\lambda}[p_l - c_2^{s_1}(p_h, p_l) + \psi(c_2^{s_1}(p_h, p_l))] \Big] x_2^{s_1}(p_h, p_l)
$$

$$
+ \alpha(1-\alpha) \Big[ Q_h - \frac{1}{1-\lambda}[p_l - c_1^{s_1}(p_l, p_h) + \psi(c_1^{s_1}(p_l, p_h))] \Big] x_1^{s_1}(p_l, p_h)
$$

$$
+ \alpha(1-\alpha) \Big[ Q_l - \frac{1}{1-\lambda}[p_h - c_2^{s_1}(p_l, p_h) + \psi(c_2^{s_1}(p_l, p_h))
$$

$$
+ \frac{\alpha\lambda_f}{(1-\alpha)(1+\lambda_f)}\Phi(c_2^{s_1}(p_l, p_h))] \Big] x_2^{s_1}(p_l, p_h)
$$

$$
+ (1-\alpha)^2 \Big[ Q_h - \frac{1}{1-\lambda}[p_h - c_1^{s_1}(p_h, p_h) + \psi(c_1^{s_1}(p_h, p_h)) + \frac{\alpha}{1-\alpha}\Phi(c_1^{s_1}(p_h, p_h))] \Big] x_1^{s_1}(p_h, p_h)
$$

$$
+ (1-\alpha)^2 \Big[ Q_l - \frac{1}{1-\lambda}[p_h - c_2^{s_1}(p_h, p_h) + \psi(c_2^{s_1}(p_h, p_h))
$$

$$
+ \frac{\alpha\lambda_f}{(1-\alpha)(1+\lambda_f)}\Phi(c_2^{s_1}(p_h, p_h))] \Big] x_2^{s_1}(p_h, p_h) \Bigg\}.
$$

We compare the terms comprising $x_i^{s_1}(P_1, P_2)$ and derive the following solutions by maximizing each part of (A9) with regard to $c_i^{s_1}(P_1, P_2)$: $c_1^{s_1}(p_l, p_l) = c_2^{s_1}(p_l, p_l) = c_2^{s_1}(p_h, p_l) = c_1^{s_1}(p_l, p_h) = c^*$, $c_1^{s_1}(p_h, p_l) = c_1^{s_1}(p_h, p_h) = \tilde{c}$, $c_2^{s_1}(p_l, p_h) = c_2^{s_1}(p_h, p_h) = \hat{c}$, $x_1^{s_1}(p_l, p_l) = 1$, $x_1^{s_1}(p_l, p_h) = 1$ if

$\Delta Q > \frac{1}{1-\lambda}(-\Delta p + \hat{c} - c^* - \psi(\hat{c}) + \psi(c^*) - \frac{\alpha\lambda_f}{(1-\alpha)(1+\lambda_f)}\Phi(\hat{c}))$, $x_1^{s_1}(p_h, p_l) = 1$ if $\Delta Q >$

$\frac{1}{1-\lambda}(\Delta p - \tilde{c} + c^* + \psi(\tilde{c}) - \psi(c^*) + \frac{\alpha}{1-\alpha}\Phi(\tilde{c}))$, $x_1^{s_1}(p_h, p_h) = 1$ if $\Delta Q > \frac{1}{1-\lambda}(\hat{c} - \tilde{c} +$

$\psi(\tilde{c}) - \psi(\hat{c}) + \frac{\alpha}{1-\alpha}\Phi(\tilde{c}) - \frac{\alpha\lambda_f}{(1-\alpha)(1+\lambda_f)}\Phi(\hat{c}))$, where $\tilde{c}$ is the solution of $\psi'(\tilde{c}) = 1 -$

$\frac{\alpha}{1-\alpha}\Phi'(\tilde{c})$, $\hat{c}$ is the solution of $\psi'(\hat{c}) = 1 - \frac{\alpha\lambda_f}{(1-\alpha)(1+\lambda_f)}\Phi'(\hat{c})$. We omit the nonsensical results such as $c_2^{s_1}(p_l, p_l) = c^*$, $x_2^{s_1}(p_l, p_l) = 0$ and obtain the refined solutions of $c_i^{s_1}(P_1, P_2)$. Similarly, we obtain the refined solutions of $c_i^{s_2}(P_1, P_2)$ and $c_i^{s_0}(P_1, P_2)$ in Tables A3 and A4.

We can conclude that the benefit is fixed at the highest level of $c^*$ when the PV firm is a low-price type. The benefits of a high-price firm under $s_1$ and $s_2$ decrease to $\tilde{c}$ and $\hat{c}$. A high-price firm in status $s_0$ obtains a benefit $\check{c}$, where $c^*, \tilde{c}, \hat{c}$, and $\check{c}$ are the solutions of $\psi'(c) = 1$, $\psi'(\tilde{c}) = 1 - \frac{\alpha}{1-\alpha}\Phi'(\tilde{c})$, $\psi'(\hat{c}) = 1 - \frac{\alpha\lambda_f}{(1-\alpha)(1+\lambda_f)}\Phi'(\hat{c})$, and $\psi'(\check{c}) = 1 - \frac{\alpha(2+\lambda_f)}{(1+\lambda_f)(1-\alpha)}\Phi'(\hat{c})$, respectively. $\square$

**Table A2.** Refined solutions of $c_i^{s_1}(P_1, P_2)$ at first-score $s_1$.

| $c_i^{s_1}(P_1,P_2)$ / $x_i^{s_1}(P_1,P_2)$ | $c_1^{s_1}(p_l,p_l)$ | $c_1^{s_1}(p_l,p_h)$ | $c_2^{s_1}(p_l,p_h)$ | $c_1^{s_1}(p_h,p_l)$ | $c_2^{s_1}(p_h,p_l)$ | $c_1^{s_1}(p_h,p_h)$ | $c_2^{s_1}(p_h,p_h)$ |
|---|---|---|---|---|---|---|---|
| $x_1^{s_1}(p_l,p_l)=1$ | $c^*$ | | | | | | |
| $x_1^{s_1}(p_l,p_h)=1$ * | | $c^*$ | $\hat{c}$ | | | | |
| $x_1^{s_1}(p_h,p_l)=1$ ** | | | | $\tilde{c}$ | $c^*$ | | |
| $x_1^{s_1}(p_h,p_h)=1$ *** | | | | | | $\tilde{c}$ | $\hat{c}$ |

Note: The above values are true under the following conditions. * $\Delta Q > \dfrac{1}{1-\lambda}\left(-\Delta p + \hat{c} - c^* - \psi(\hat{c}) + \psi(c^*) - \dfrac{\alpha\lambda_f}{(1-\alpha)(1+\lambda_f)}\Phi(\hat{c})\right)$. ** $\Delta Q > \dfrac{1}{1-\lambda}\left(\Delta p - \tilde{c} + c^* + \psi(\tilde{c}) - \psi(c^*) + \dfrac{\alpha}{1-\alpha}\Phi(\tilde{c})\right)$. *** $\Delta Q > \dfrac{1}{1-\lambda}\left(\hat{c} - \tilde{c} + \psi(\tilde{c}) - \psi(\hat{c}) + \dfrac{\alpha}{1-\alpha}\Phi(\tilde{c}) - \dfrac{\alpha\lambda_f}{(1-\alpha)(1+\lambda_f)}\Phi(\hat{c})\right)$.

**Table A3.** Refined solutions of $c_i^{s_2}(P_1, P_2)$ at first-score $s_2$.

| $c_i^{s_2}(P_1,P_2)$ / $x_i^{s_2}(P_1,P_2)$ | $c_2^{s_2}(p_l,p_l)$ | $c_1^{s_2}(p_l,p_h)$ | $c_2^{s_2}(p_l,p_h)$ | $c_2^{s_2}(p_h,p_l)$ | $c_1^{s_2}(p_h,p_l)$ | $c_1^{s_2}(p_h,p_h)$ | $c_2^{s_2}(p_h,p_h)$ |
|---|---|---|---|---|---|---|---|
| $x_2^{s_2}(p_l,p_l)=1$ | $c^*$ | | | | | | |
| $x_2^{s_2}(p_l,p_h)=1$ * | | $c^*$ | $\tilde{c}$ | | | | |
| $x_2^{s_2}(p_h,p_l)=1$ ** | | | | $c^*$ | $\hat{c}$ | | |
| $x_2^{s_2}(p_h,p_h)=1$ *** | | | | | | $\hat{c}$ | $\tilde{c}$ |

Note: The above values are true under the following conditions. * $\Delta Q > \dfrac{1}{1-\lambda}\left(-\Delta p + \hat{c} - c^* - \psi(\hat{c}) + \psi(c^*) - \dfrac{\alpha\lambda_f}{(1-\alpha)(1+\lambda_f)}\Phi(\hat{c})\right)$. ** $\Delta Q > \dfrac{1}{1-\lambda}\left(\Delta p - \tilde{c} + c^* + \psi(\tilde{c}) - \psi(c^*) + \dfrac{\alpha}{1-\alpha}\Phi(\tilde{c})\right)$. *** $\Delta Q > \dfrac{1}{1-\lambda}\left(\hat{c} - \tilde{c} + \psi(\tilde{c}) - \psi(\hat{c}) + \dfrac{\alpha}{1-\alpha}\Phi(\tilde{c}) - \dfrac{\alpha\lambda_f}{(1-\alpha)(1+\lambda_f)}\Phi(\hat{c})\right)$.

**Table A4.** Refined solutions of $c_i^{s_0}(P_1, P_2)$ at first-score $s_0$.

| $c_i^{s_0}(P_1,P_2)$ / $x_i^{s_0}(P_1,P_2)$ | $c_1^{s_0}(p_l,p_l)$ | $c_2^{s_0}(p_l,p_l)$ | $c_1^{s_0}(p_l,p_h)$ | $c_2^{s_0}(p_h,p_l)$ | $c_1^{s_0}(p_h,p_h)$ | $c_2^{s_0}(p_h,p_h)$ |
|---|---|---|---|---|---|---|
| $x_1^{s_0}(p_l,p_l)=\frac{1}{2}$ | $c^*$ | $c^*$ | | | | |
| $x_1^{s_0}(p_l,p_h)=1$ | | | $c^*$ | | | |
| $x_2^{s_0}(p_h,p_l)=1$ | | | | $c^*$ | | |
| $x_2^{s_0}(p_h,p_h)=\frac{1}{2}$ | | | | | $\check{c}$ | $\check{c}$ |

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
