# Peer review of "When Will First-Price Work Well? The Impact of Anti-Corruption Rules on Photovoltaic Power Generation Procurement Auctions"

_sustainability, doi:10.3390/su15043441_

Round 1
Reviewer 1 Report
This study considers a corruption-proof environment wherein corruption is strictly suppressed. It elaborates a mechanism designed to explore the impact of corruption-proof measures on PV procurement auctions through adverse selection related to the revelation principle. I think the paper is rather interesting. I have some suggestions for large items to tackle (along with some minor gripes) that should be addressed before publication in order to make the paper more useful to other practitioners in the field.
1. Major issues:
(1) The Abstract is the epitome of the whole paper, which reflects the theme, purpose and the main/core findings (not all the findings) of the paper. Therefore, the Abstract needs to be improved.
(2) There is a lack of clarity in the logic and structure. It is suggested that simplification and conciseness should be closely related to the content of each paragraph. Make sure that the overall logic is rigorous and clear. There should be a clear internal logic within each paragraph in which methodology, content, and findings can be discussed one by one.
(3) Literature reviews should be focused on the research question or purpose of the paper. It is not necessary to include non-relevant literature. Additionally, literature that is not closely related to the research content can be omitted.
2. Minor issues:
(1) In Introduction: Summarizing different kind of corrupt practice PV is strongly recommended to add.
(2) In Figure 4-10, the information mining behind the data needs to be further strengthened.
(3) In discussion: Key issues to be discussed should be clear, don't simply repeat the conclusion in previous chapters.
Author Response
Response to Reviewer 1 Comments
Point 1: Q1: 1. Major issues:(1)The Abstract is the epitome of the whole paper, which reflects the theme, purpose and the main/core findings (not all the findings) of the paper. Therefore, the Abstract needs to be improved.
Response 1: Thank you for your valuable comment for improving our manuscript. We replace sentences with appropriate descriptions. Parts of the revisions are presented as follows.
Page 1, Abstract:
It elaborates a mechanism to explore the impact of corruption-proof measures on PV procurement auctions. It adopts incentive compatible constraints based on revelation principle to reflect PV firms' optimal utilities. It elaborates a mechanism designed to explore the impact of corruption-proof measures on PV procurement auctions through adverse selection related to the revelation principle.
Point 2: Q2: 1. Major issues:(2)There is a lack of clarity in the logic and structure. It is suggested that simplification and conciseness should be closely related to the content of each paragraph. Make sure that the overall logic is rigorous and clear. There should be a clear internal logic within each paragraph in which methodology, content, and findings can be discussed one by one.
Response 2: Thank you for your valuable suggestion. We made many changes to ensure that each paragraph was concise. To this end, we optimized the overall logic to ensure that it was rigorous and clear. We adjust the structure of the paper, such as put the issues solved by the research and the innovations after the related literature, so as to make the paper more logical. Parts of the revisions are presented as follows.
Page 4, Section 1:
This study focuses on the influence of corruption-proof measures on PV procurement auctions. … Therefore, this study addresses the following issues:
Q1. What are the advantages and disadvantages of the first-price concept for PV procurement auctions under a strict corruption-proof environment?
Q2. What strategy, first-price or first-score, should regulators employ to maximize social welfare?
Point 3: Q3: Major issues:(3)Literature reviews should be focused on the research question or purpose of the paper. It is not necessary to include non-relevant literature. Additionally, literature that is not closely related to the research content can be omitted.
Response 3: We express our sincere gratitude for the friendly criticism. We omit the irrelevant literature to make the paper clearer. Parts of the revisions are presented as follows.
Page 3, Section 1:
For a classical auction-corruption-related theory, studies on bid readjustment are abundant, where the auctioneer favors one of the bidders and readjusts its price bid in exchange for corruption. … Although studies provide some valuable insights, an implausible feature of these approaches is that, in most countries, bid readjustment is illegal and cannot be executed in external oversight.
Point 4: Q4: Minor issues: (1)In Introduction: Summarizing different kind of corrupt practice PV is strongly recommended to add.
Response 4: Thank you for your valuable comment for improving our manuscript. We summarize two corrupt PV practices. Parts of the revisions are presented as follows.
Page 1, Section 0:
There are two kinds of corruption, vertical corruption and horizontal corruption. … The other widely prevalent form of a collusive bid is bid rigging, namely, horizontal corruption. It arises when a subset, or possibly all, of the bidders, acts collusively and engages in bid rigging with a view to obtaining higher prices in a procurement auction.
Point 5: Q5: Minor issues: (2)In Figure 4-10, the information mining behind the data needs to be further strengthened.
Response 5: Thank you very much for the insightful advice. We explain the data information of Figure 4-10. Parts of the revisions are presented as follows.
Page 10, Section 3:
Figure 4(a) illustrates the relationship between social welfare and the upper and lower boundaries, and , of PV enterprise quality. Figure 4(b) illustrates the relationship between social welfare and the upper and lower boundaries, and , of PV enterprise price.
Point 6: Q6: Minor issues: (3)In discussion: Key issues to be discussed should be clear, don't simply repeat the conclusion in previous chapters.
Response 6: Thank you for pointing this out. The following problems are discussed in this discussion: firstly, the results of this paper are compared with those of the predecessors. Secondly, some contributions are highlighted from the perspective of empirical research results. Third, in the context of anti-corruption and the impact of photovoltaic power generation, we have made some contributions to future research. Fourthly, the application of this research is carried out.
Page 14, Section 4:
Comparing our results with those of previous studies leads to the following findings. …In conclusion, our results are consistent with those of auction comparison and illustrate some novel insights.
Page 15, Section 4:
This study has some contributions in terms of empirical research results. We highlight the contributions as follows. … As a result, regulators should employ a first-score mechanism to maximize social welfare, especially in a high-quality and low-price PV market.
This study contributes to future research in the context of anti-corruption and the impact of PV power generation. … Future anti-corruption research can use this mechanism as the baseline of anti-corruption model framework.
Page 16, Section 4:
This study can be applied to procurement auctions in other renewable energy projects. … Therefore, our study is only applicable for the renewable power generation station projects.

Reviewer 2 Report
In this study, the authors consider a corruption-proof environment where corruption is strictly suppressed. They elaborates a mechanism for examining the impact of corruption-proof measures related to the revelation principle on photovoltaic procurement auctions. First-price auctions and first-score auctions provide a description of market outcomes using the Bayesian Nash equilibrium. The study is very interesting. And the design of the study is clearly presented and the methods are adequately described. This article deserves to be published, and there are a few questions for the authors to consider before publication.
Q1: Is this study only applicable to photovoltaic power generation procurement auctions, or can it be applied to auctions in other fields, and can it be discussed in the discussion section?
Q2: The text in all the figures is a little small, and can be enlarged appropriately.
Q3: The conclusion section of the article is missing.
Author Response
Response to Reviewer 2 Comments
Point 1: Q1: Is this study only applicable to photovoltaic power generation procurement auctions, or can it be applied to auctions in other fields, and can it be discussed in the discussion section?
Response 1: Thank you for your valuable comment for improving our manuscript. This study can be applied to procurement auctions in other renewable energy projects. We talked about this in detail in the discussion section of the updated version. Parts of the revisions are presented as follows.
Page 15, Section 4:
This study can be applied to procurement auctions in other renewable energy projects. … Therefore, our study is only applicable for the renewable power generation station projects.
Point 2: Q2: The text in all the figures is a little small, and can be enlarged appropriately.
Response 2: Thank you for your valuable suggestion. We have enlarged all the images to make sure the text is properly enlarged and clear. Parts of the revisions are presented as follows.
Page 10, Section 3:
(a) (b)
Figure 4. Social welfare on quality and price (a) social welfare on quality, (b) social welfare on price.
Point 3: Q3: The conclusion section of the article is missing.
Response 3: We express our sincere gratitude for the friendly criticism. We added section 5. Conclusion in the final part of the body. Parts of the revisions are presented as follows.
Page 16, Section 5:
In summary, our study is innovative in several ways. Firstly, we arrange the content according to the findings of literature research. … To this end, we have developed two auction markets, high-quality-low-price and low-quality-high-price, and formulated policies to curb PV corruption.
Secondly, by comparing the first-price auction with the first-score auction, we find out the appropriate strategy to restrain the corruption of PV procurement auction. … In equilibrium analysis, the first-price and first-score of PV procurement auctions in social welfare, utility, corruption benefit and total income are compared.
Finally, we use Bayesian Nash equilibrium, revelation principle and exogenous favoritism to arrange the study. … Therefore, the application of exogenous favoritism reflects the reality of auction practice.

Reviewer 3 Report
1. The authors should compare the current results with the previous studies. Try to highlight the contribution in terms of empirical findings.
2. How will this research contribute to future research in the context of the impact of anticorruption and photovoltaic power generation?
3. The conclusion should be added after the discussion section. 4. Also need to add some more comments in the discussion section.
Author Response
Response to Reviewer 3 Comments
Point 1: Q1: 1. The authors should compare the current results with the previous studies. Try to highlight the contribution in terms of empirical findings.
Response 1: Thank you for your valuable comment for improving our manuscript. We added a comparison between the current results and the results of previous studies. We also added a highlight discribing the contribution in terms of empirical findings. We talked about these in detail in the discussion section of the updated version. Parts of the revisions are presented as follows.
Page 14, Section 4:
Comparing our results with those of previous studies leads to the following findings. … In conclusion, our results are consistent with those of auction comparison and illustrate some novel insights.
Page 15, Section 4:
This study has some contributions in terms of empirical research results. We highlight the contributions as follows. … As a result, regulators should employ a first-score mechanism to maximize social welfare, especially in a high-quality and low-price PV market.
Point 2: Q2: 2. How will this research contribute to future research in the context of the impact of anticorruption and photovoltaic power generation?
Response 2: Thank you for your valuable suggestion. We have added contributions to future research in the context of anti-corruption and the impact of PV power generation. Parts of the revisions are presented as follows.
Page 15, Section 4:
This study contributes to future research in the context of anti-corruption and the impact of PV power generation. ... Future anti-corruption research can use this mechanism as the baseline of anti-corruption model framework.
Point 3: Q3: 3. The conclusion should be added after the discussion section.
Response 3: We express our sincere gratitude for the friendly criticism. We added section 5. Conclusion in the final part of the body. Parts of the revisions are presented as follows.
Page 16, Section 5:
In summary, our study is innovative in several ways. Firstly, we arrange the content according to the findings of literature research. … To this end, we have developed two auction markets, high-quality-low-price and low-quality-high-price, and formulated policies to curb PV corruption.
Secondly, by comparing the first-price auction with the first-score auction, we find out the appropriate strategy to restrain the corruption of PV procurement auction. … In equilibrium analysis, the first-price and first-score of PV procurement auctions in social welfare, utility, corruption benefit and total income are compared.
Finally, we use Bayesian Nash equilibrium, revelation principle and exogenous favoritism to arrange the study. … Therefore, the application of exogenous favoritism reflects the reality of auction practice.
Point 3: Q4: 4. Also need to add some more comments in the discussion section.
Response 4: Thank you for your valuable comment for improving our manuscript. This study can be applied to procurement auctions in other renewable energy projects. We add these comments in detail in the discussion section of the updated version. Parts of the revisions are presented as follows.
Page 16, Section 4:
This study can be applied to procurement auctions in other renewable energy projects. … Therefore, our study is only applicable for the renewable power generation station projects.

Reviewer 4 Report
(1) Some abbreviations appear in their full names only in the abstract. In fact, abbreviations should be written in full when they first appear in the abstract and body of the paper.
(2) According to the research innovation points mentioned in the introduction, 1) Why Bayesian Nash equilibrium is adopted and what are its advantages compared with general equilibrium;2) Explain the necessity of using the disclosure principle and adding incentive compatibility and individual rationality constraints to solve the Bayesian Nash equilibrium.
(3) It is suggested to adjust the structure of the paper, such as put the issues solved by the research and the innovations after the related literature, so as to make the paper more logical.
(4) There are unexplained parameters, such as m mentioned in the related literature.
(5) As mentioned in the literature, endogenous favoritism is considered in this study, compared with the recent first-score-related studies considering exogenous favoritism. What is the advantage of this design?
(6) Figure title format is not standard, and it should be centered below the picture.
(7) There is a problem with the use of font italics in the model section。
(8) Paragraph indentation formatting is inconsistent.
(9) Chapter titles are too simple or even wrong. For example, the title analyses in Chapter 3 should be analysis. It is suggested to optimize the title based on specific content.
(10) It is suggested to adjust the layout of the model to enhance readability. In addition, it is suggested to optimize some charts to make them more beautiful.
Author Response
Response to Reviewer 4 Comments
Point 1: Q1: (1) Some abbreviations appear in their full names only in the abstract. In fact, abbreviations should be written in full when they first appear in the abstract and body of the paper.
Response 1: Thank you for your valuable comment for improving our manuscript. We add an abbreviation for photovoltaic in the body of the paper. Parts of the revisions are presented as follows.
Page 1, Section 0:
Bidding corruption is extensively distributed in the current procurement of solar photovoltaic (PV) plant projects.
Point 2: Q2: (2) According to the research innovation points mentioned in the introduction, 1) Why Bayesian Nash equilibrium is adopted and what are its advantages compared with general equilibrium; 2) Explain the necessity of using the disclosure principle and adding incentive compatibility and individual rationality constraints to solve the Bayesian Nash equilibrium.
Response 2: Thank you for your valuable suggestion. We explain the use of Bayesian Nash equilibrium as a research problem and explain its advantages over general equilibrium. At the same time, we explain the necessity to apply the principle of revelation, incentive compatibility and individual rational constraint to solve the Bayesian Nash equilibrium. Parts of the revisions are presented as follows.
Page 2, Section 0:
The Bayesian Nash equilibrium is adopted as the research topic and the principle of revelation is adopted as the method to solve this problem. The reasons are as follows. Firstly, Nash equilibrium solves the problem of game, while general equilibrium problem deals with the problem of program. … Secondly, we adopt the revelation principle to solve the Bayesian Nash equilibrium, because it can make PV enterprises show the true types. … To this end, we add IC constraints to make PV enterprises tell the truth, and IR constraints to ensure that PV enterprises participate in the game.
Point 3: Q3: (3) It is suggested to adjust the structure of the paper, such as put the issues solved by the research and the innovations after the related literature, so as to make the paper more logical.
Response 3: We express our sincere gratitude for the friendly criticism. We put the issues solved by the research and the innovations after the related literature. Parts of the revisions are presented as follows.
Page 4, Section 1:
This study focuses on the influence of corruption-proof measures on PV procurement auctions. … Therefore, this study addresses the following issues:
Q1. What are the advantages and disadvantages of the first-price concept for PV procurement auctions under a strict corruption-proof environment?
Q2. What strategy, first-price or first-score, should regulators employ to maximize social welfare?
Point 4: Q4: (4) There are unexplained parameters, such as m mentioned in the related literature.
Response 4: Thank you for your valuable comment for improving our manuscript. We add explanations to the unexplained parameters. Parts of the revisions are presented as follows.
Page 3, Section 1:
If , then the agent favors Firm 1 by exaggerating its quality by a multiplier , as long as Firm 1 wins with the manipulation.
Point 5: Q5: (5) As mentioned in the literature, endogenous favoritism is considered in this study, compared with the recent first-score-related studies considering exogenous favoritism. What is the advantage of this design?
Response 5: Thank you very much for the insightful advice. We compare the endogenous favoritism with the exogenous favoritism and get the advantage of the endogenous favoritism. Parts of the revisions are presented as follows.
Page 3, Section 1:
The advantage of exogenous favoritism is that it reflects the reality of procurement auctions. ... Therefore, we use exogenous favoritism to study the establishment of the models.
Point 6: Q6: (6) Figure title format is not standard, and it should be centered below the picture.
Response 6: Thank you for pointing this out. We apologize for ignoring the Figure title format. We have corrected the formatting of the caption to make it standard and centered. Parts of the revisions are presented as follows.
Page 10, Section 3:
(a)
Figure 4. Social welfare on quality and price (a) social welfare on quality, (b) social welfare on price.
Point 7: Q7: (7) There is a problem with the use of font italics in the model section.
Response 7: Thank you for pointing this out. We apologize for the inappropriate use of italics in the model section. We remove the italics and made the model standard. Parts of the revisions are presented as follows.
Page 7, Section 2:
Point 8: Q8: (8) Paragraph indentation formatting is inconsistent.
Response 8: Thank you for pointing this out. We apologize for the inappropriate use of paragraph indentation. We correct the mistake and establish the indentation standard. Parts of the revisions are presented as follows.
Page 8, Section 2:
Point 9: Q9: (9) Chapter titles are too simple or even wrong. For example, the title analyses in Chapter 3 should be analysis. It is suggested to optimize the title based on specific content.
Response 9: Thank you for your valuable comments for improving our manuscript. We optimize the title based on specific content. Parts of the revisions are presented as follows.
Page 9, Section 3:
- Equilibrium analysis on model's outcome
Page 10, Section 3:
3.2 Corruption benefit of PV firm
Point 10: Q10: (10) It is suggested to adjust the layout of the model to enhance readability. In addition, it is suggested to optimize some charts to make them more beautiful.
Response 10: Thank you for your valuable comments and suggestions for improving our manuscript. We adjust the layout of the model to enhance readability. In addition, we optimize some charts to make them more beautiful. Parts of the revisions are presented as follows.
Page 7, Section 3:
Page 4, Section 2:
Figure 1. Business model of a PV procurement auction
